# Single-cell transcriptional changes associated with drug tolerance and response to combination therapies in cancer

Alexandre F. Aissa [1], Abul B. M. M. K. Islam [1,2], Majd M. Ariss [1], Cammille C. Go[1], Alexandra E. Rader[1], Ryan D. Conrardy[1], Alexa M. Gajda[1], Carlota Rubio-Perez [3], Klara Valyi-Nagy[4], Mary Pasquinelli[5], Lawrence E. Feldman [5], Stefan J. Green [6], Nuria Lopez-Bigas[3], Maxim V. Frolov [1] & Elizaveta V. Benevolenskaya [1]✉

Tyrosine kinase inhibitors were found to be clinically effective for treatment of patients with certain subsets of cancers carrying somatic mutations in receptor tyrosine kinases. However, the duration of clinical response is often limited, and patients ultimately develop drug resistance. Here, we use single-cell RNA sequencing to demonstrate the existence of multiple cancer cell subpopulations within cell lines, xenograft tumors and patient tumors. These subpopulations exhibit epigenetic changes and differential therapeutic sensitivity. Recurrently overrepresented ontologies in genes that are differentially expressed between drug tolerant cell populations and drug sensitive cells include epithelial-to-mesenchymal transition, epithelium development, vesicle mediated transport, drug metabolism and cholesterol homeostasis. We show analysis of identified markers using the LINCS database to predict and functionally validate small molecules that target selected drug tolerant cell populations. In combination with EGFR inhibitors, crizotinib inhibits the emergence of a defined subset of EGFR inhibitor-tolerant clones. In this study, we describe the spectrum of changes associated with drug tolerance and inhibition of specific tolerant cell subpopulations with combination agents.

[1] Department of Biochemistry and Molecular Genetics, University of Illinois at Chicago, Chicago, IL, USA. [2] Department of Genetic Engineering and Biotechnology, University of Dhaka, Dhaka, Bangladesh. [3] Biomedical Genomics Lab, Institute for Research in Biomedicine (IRB), Barcelona, Spain. [4] Department of Pathology, University of Illinois at Chicago, Chicago, IL, USA. [5] Department of Medicine, Section of Hematology/Oncology, University of Illinois at Chicago, Chicago, IL, USA. [6] Genome Research Core, Research Resources Center, University of Illinois at Chicago, Chicago, IL, USA. ✉email: evb@uic.edu

The identification of actionable mutations in human tumors has dramatically altered cancer management. One of the most successful examples of targeted therapy is the implementation of tyrosine kinase inhibitors (TKIs) as first-line therapy in patients with activating mutations in the receptor tyrosine kinase gene EGF receptor (EGFR)[1–3]. Despite the fact that a majority of patients show good initial responses to treatment with the EGFR-TKIs erlotinib and gefitinib, they frequently develop resistance. Additionally, there are patients whose cancers are immediately refractory.

Two mechanisms underlying intrinsic and acquired drug resistance are being explored to optimize the clinical utility of TKIs[4–6]. The first group is related to the targeted kinase and includes secondary mutations that attenuate drug inhibition. This group is exemplified by EGFR mutations T790M and C797S, which may either pre-exist in treatment-naïve tumors or arise de novo following drug exposure[7]. The second resistance mechanism converges on signaling pathways which share effectors with the receptor tyrosine kinase (RTK) or those downstream pathways to bypass the targeted RTK[8,9]. MET- or IGF-1R-dependent maintenance of PI3K-AKT-mTOR signaling, BRAF or RAS-mediated triggering of the MAPK pathway, activation of FGFR1, HER2, or AXL RTKs, all serve as alternate routes for reactivation of signaling downstream of the inhibited RTK.

Significant variability in drug response at the level of individual cells within a clonal cell population has been observed in multiple contexts[10]. Consequently, the heterogeneous response of single-cell-derived persisters to anticancer therapies has been noted for many anticancer drugs[11]. The observation that drug-resistant colonies derived from a single cell were deficient in many known erlotinib-resistance mechanisms, including epithelial-to-mesenchymal transition (EMT), activation of nuclear factor NF-kappa-B (NF-κB), insulin-like growth factor 1 receptor (IGF-1R), and AXL, was quite surprising and suggested that different surviving cells employ distinct mechanisms. In fact, immunohistochemical staining and fluorescent in situ hybridization (FISH) in human tissues revealed a wide variation in expression of multiple cancer biomarkers between different cells. These routine observations were supported by more advanced methods, such as immunological assays with antibodies revealing pathway activation, functional assays with small-molecule inhibitors, and whole-exome sequencing of matched patient's samples. As genotyping and drug-sensitivity testing required high cell numbers, which usually could only be achieved after 12–16 weeks of expanding rare tolerant cells in culture, there was gap in analysis of genetic and epigenetic heterogeneity during the initial robust response to targeted therapy. While the identified changes were undoubtedly highly clinically relevant on the patient timescale, exclusion of information regarding the earlier sequence of events has precluded identification of combination agents that would reverse drug tolerance. Most recent approaches have produced convincing data suggesting that the complexity of resistant cells is largely underestimated. Tracking down alterations in non-small-cell lung carcinoma (NSCLC) cell line PC9 by next-generation sequencing (NGS), droplet digital PCR (ddPCR), and tagging individual cells with unique barcodes, showed that they were associated with pre-existing resistant clones[7,12,13].

In this study, we build on the idea that cell-to-cell differences are critical to therapeutic response to single-agent therapies and that revelation of early phenotypic transitions may offer new combination agents in patients with EGFR-mutated NSCLC. To better understand the differences in the response of individual cells to drug treatment, we apply methods for precise and comprehensive single-cell analysis[14] using an established preclinical model of NSCLC that responds to tyrosine kinase inhibitors such as erlotinib. We perform treatment of cells grown in cell culture or mouse xenografts with EGFR-TKIs followed by single-cell RNA sequencing (scRNA-seq). Integrative data analysis uncovers mechanisms through which drug tolerance arises in NSCLC cell line models during treatment. This pertains to the establishment of distinct drug tolerant states that can co-occur within NSCLC cell populations and express distinct combinations of markers that can ultimately be used as prognostic and/or therapeutic targets for small-molecule therapies.

## Results

**Discovery of drug-tolerant states in PC9 cells treated with erlotinib.** First-generation inhibitors such as erlotinib have revolutionized the treatment of EGFR-mutant NSCLCs. PC9 cells contain an exon 19 deletion (ΔE746-A750, called EGFRex19 hereafter) in EGFR gene and exemplify changes in patient tumors associated with intrinsic and acquired TKI resistance[15]. Erlotinib and osimertinib, an irreversible third-generation EGFR TKI that is now is used as first-line treatment for patients with EGFR mutation-positive NSCLC[16,17], are effective on PC9 at low nanomolar concentrations (Fig. 1a). Erlotinib exerts cytostatic and cytotoxic effects on PC9 at 2 μM, the concentration achieved in patients receiving standard therapy[18]. However, after continuous treatment with the erlotinib some subpopulations of cells survive and begin expansion (Fig. 1b and Supplementary Fig. 1a). Such resistance is clinically relevant to NSCLC patients that were treated with EGFR inhibitors[6]. Even the earliest drug-tolerant persisters (DTPs) and drug-tolerant expanded persisters (DTEPs)[15,19] are tolerant to much higher erlotinib concentrations than the original PC9 cells (Fig. 1c). One of the mechanisms explaining the emergence of eventually resistant clones was attributed to the T790M "gatekeeper" mutation in EGFR, which reportedly pre-exists or develops after several months of continuous treatment[7]. We confirmed, consistent with previous reports[7,15], that the T790M mutation was not enriched in the initial emerging PC9 DTEPs as its frequency remained at around 0.2% at Day 11 of treatment (Supplementary Fig. 1b).

We considered that earlier events in response to EGFR TKI involve epigenetic mechanisms. This idea is supported by limited expansion of DTPs after application of epigenetic inhibitors[15,20,21]. Therefore, we sought comprehensively characterize drug-tolerant states in PC9 cells that were treated with erlotinib for a relatively short time (Fig. 1d). Traditional methods were unfeasible due to small cell numbers. So, we analyzed 848 PC9 cells subjected to consecutive erlotinib treatment (for 1, 2, 4, 9, and 11 days) and 756 control cells using Drop-seq[14] (Supplementary Tables 1 and 2 and Supplementary Fig. 1c).

We used Seurat's Uniform Manifold Approximation and Projection (UMAP) dimensionality reduction[22] that allows for visualization of cells with similar gene expression signatures and principal component (PC) loadings, based on nearness to each other in the embedding. We used clustering to define cell populations with their associated respective markers, and considered several parameters described in Methods, including the mean silhouette width[23], as a measure of stability of cluster assignment and separation from neighboring clusters. The untreated cells (D0) were represented by three clusters, whereas the erlotinib-treated cells were represented by five clusters that spread out based on the number of consecutive days following treatment (Fig. 1e and Supplementary Fig. 1d, e). Inferring cell fate progression over time using Velocyto software[24] confirmed cluster directionality (Fig. 1e). As expected[10,15], DTPs withdrew from the cell cycle and entered a quiescent state, while DTEPs became proliferation-competent (Fig. 1e). Regressing out cell cycle genes returned similar clusters (Supplementary Fig. 1f, g), indicating that

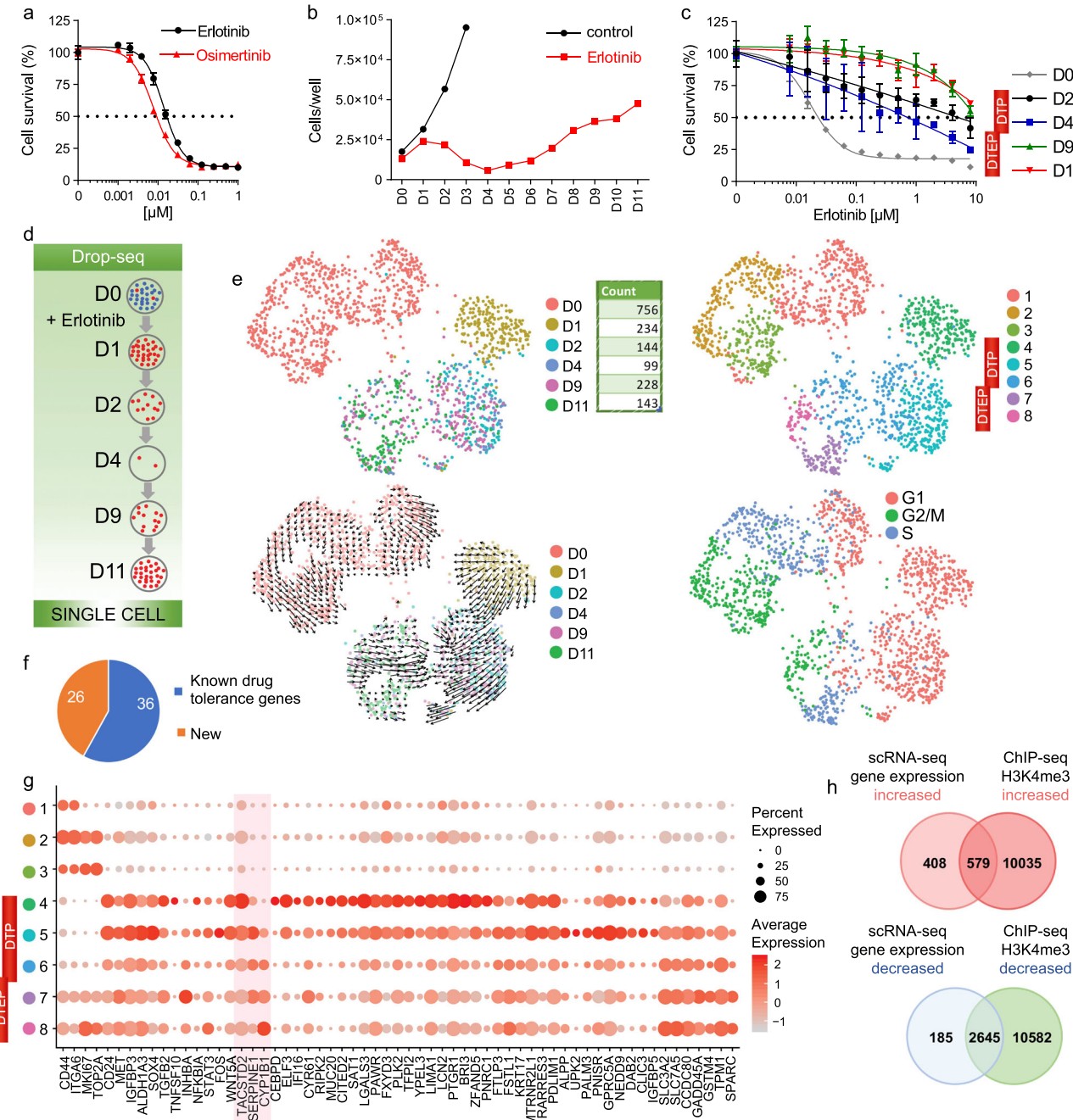

**Fig. 1 Drop-seq recapitulates diversity of drug-tolerant states. a** Dose response of PC9 cells to erlotinib and osimertinib at day 3 of treatment. Cell counting was performed using Hoechst. Mean ± standard deviation (SD) for $n = 3$ replicate wells are shown. **b** Growth curve of PC9 cells treated with erlotinib (2 μM) during 11 days (D0–D11), and growth curve of PC9 cells without addition of erlotinib during 3 days (D1–D3). Cell counting was performed using hemocytometer, and data represents mean values for $n = 2$ replicate wells. **c** Dose response to erlotinib, describing PC9 cells (D0), and the DTPs and DTEPs generated from the original PC9 cells by treating in 2 μM erlotinib for respective number of days (D2, D4, D9, and D11). D0, D2, D4, D9, and D11 PC9 cells were treated with erlotinib dilution series for 3 days, after which cells were counted using Hoechst, mean ± SD for $n = 3$ replicate wells. **d** Schematic for the set of consecutive samples for single-cell RNA-seq. D0 are untreated cells, and D1 through D11 is the duration of treatment with the drug. Drop-seq was done on cells at Day 0 (D0), Day 1 (D1), Day 2 (D2), Day 4 (D4), Day 9 (D9), and Day 11 (D11) of the treatments. **e** UMAP representation of PC9 cells colored by days of treatment (left top panel), clusters (right top panel), days of treatment with Velocyto projection (left bottom panel), or the cell cycle phase positioning of each cell (right bottom panel). **f** Literature search with the top markers of tolerant states identified by Drop-seq using the terms [GENE NAME] + chemoresistance, [GENE NAME] + drug, and [GENE NAME] + resistance. **g** Dot plot of transcript expression for top cluster markers. The color of each dot represents the average expression level from low (gray) to high (red), and the size of each dot represents the percentage of the cells expressing the gene. Markers selected for functional validation are highlighted in red. **h** Overlap of genes with increased and decreased expression level and H3K4me3 level in PC9 cells treated with erlotinib for 11 days versus untreated.

cell cycle signature did not drive the distribution. The genes that Seurat[22] determined to be preferentially expressed in drug-tolerant (DT) cell populations (full list in Supplementary Data 1, and 62 representative genes in Fig. 1f, g) are subsequently referred to as DT markers. Previously described states of DTP and DTEP are shown in relation to the newly identified DT states 4 through 8 in Fig. 1g. The PubMed literature search of 62 DT markers found references to 36 genes associated with drug tolerance. Additional samples of cells treated for 11 days (D11) or untreated (D0) cells were analyzed by Drop-seq, bulk RNA-seq, and by ChIP-seq using antibodies to the histone modification, which allowed for identification of DT markers with significant difference in H3K4me3 enrichment (Fig. 1h, Supplementary Fig. 1h, i, and Supplementary Data 4–6). A majority of the differentially enriched H3K4me3 peaks that were annotated to markers' locations were positioned at TSS regions (Supplementary Data 4), and markers were commonly associated with increased H3K4me3 (Supplementary Fig. 1i), as expected for upregulated genes.

To validate markers identified by Drop-seq, we performed single-molecule RNA in situ hybridization (smRNA-FISH) and immunofluorescence. A number of long non-coding RNAs (lncRNAs) were upregulated upon erlotinib treatment, including Nuclear Paraspeckle Assembly Transcript 1 (*NEAT1*) and the Metastasis-Associated Lung Adenocarcinoma Transcription 1 *MALAT1*/*NEAT2* (Supplementary Data 1 and Supplementary Fig. 2a). For the coding RNAs, we have chosen to detect three of the most differentially expressed genes (Fig. 1g and Supplementary Fig 2a, b): the tumor-associated calcium signal transducer 2, *TACSTD2*, which is an EpCAM paralog and a transmembrane glycoprotein that plays a role in stabilization of tight junction proteins and in TGFβ signaling; *SERPINE1*/*PAI1*, a protease with several potential oncogenic roles[25,26]; and *CYP1B1*, a member of cytochrome P450 enzymes capable of metabolizing erlotinib[27]. *NEAT1*- and *MALAT1*-specific probes showed that the levels of these lncRNAs increased after erlotinib treatment (Supplementary Fig 2c, d). In scRNA-seq data, *MALAT1* yielded the highest fraction of counts in each single cell, across all sequenced cells. smFISH *MALAT1* probes, however, were much less sensitive in detecting the RNA. The number of *TACSTD2* transcripts increased after the erlotinib treatment, which was consistent with the Seurat data (Supplementary Fig. 2a–d). Immunostaining with antibodies to TACSTD2, SERPINE1, and CYP1B1 showed high increase in the level of each protein in cells that were treated with erlotinib for the time when the genes were induced in scRNA-seq data (Supplementary Fig. 2e, f). *TACSTD2* and *SERPINE1* showed induction in their relative level in RT-qPCR data (Supplementary Fig. 2g), confirming the scRNA-seq result. For other markers that changed their expression in the majority of cells as shown by scRNA-seq (Fig. 1g), we also confirmed upregulation using RT-qPCR analysis (Supplementary Fig. 2h). We used the data from the Cancer Cell Line Encyclopedia (CCLE)[28,29] to estimate the correlation of transcript levels with the protein expression levels across all markers. Gene set enrichment analysis (GSEA) showed that markers of earlier DT states (Supplementary Data 1) had very high transcript to protein correlation (Supplementary Fig. 2I and Supplementary Data 7 and 8).

We have validated that Drop-seq enables separation of not only irrelevant cell types but also cell progenitors within a cell population (Supplementary Figs. 3 and 4 and Supplementary Data 9). Within the set of six consecutive samples (Fig. 1d), DTPs and DETPs appeared as five separate clusters, 4 through 8 (Fig. 1e). The cells treated with erlotinib for one day, D1, formed a single cluster (Cluster 4). D1 were most distant from other populations and had a very high number of markers, 471 genes (Supplementary Data 1). Top Cluster 4 markers were expressed at

a lower level in untreated cells, i.e. Clusters, 1, 2, and 3 (Fig. 1g); this includes *TACSTD2* whose expression has increased in almost every surviving cell at D1 compared to untreated cells (Supplementary Figs. 1e and 2c, d). These findings make it unlikely that the Cluster 4 represents a mixture of tolerant cells and the cells at the original, sensitive state, and suggest that Cluster 4 cells are rather positioned at a transitional state to DTPs. Ranking of genes that are expressed in a majority cells of each cluster or state distinguished *CD24*, *MET*, *IGFBP3*, *ALDH1A3*, *SOX4*, *SERPINE1*, and *GPRC5A* as Cluster 5 markers, and *TPM1* as a Cluster 6 marker (Fig. 1g). Cluster 7 was characterized by expression of *INHBA*, and Cluster 8 preferentially expressed *CYP1B1*, *SLC3A2*, and *SLC7A5* (Fig. 1g). Because the surviving cells pass through five different states rather than the fixed state of DTP before becoming fully tolerant, we refer to them as drug-tolerant (DT) states.

**Reversible drug-tolerant states**. Drug tolerance may be a transient state, which has been clinically exemplified by the "drug holiday" phenomenon with EGFR inhibitors[30]. It is unknown whether similar or different cell populations emerge when the cells are removed from a drug treatment and then re-treated compared to continuous drug selection. We used Drop-seq to profile PC9 cells that were withdrawn from erlotinib for 6 days and then treated again for 2 days (Fig. 2a and Supplementary Fig. 5a). The cells that were re-treated after the drug holiday (+Erl, after holiday) showed a high overlap on the UMAP with the cells just before the drug holiday (+Erl, before holiday), forming Clusters 6, 7, and 8, and were distinctly positioned from the cells that were still on drug holiday, Clusters 4 and 5 (Fig. 2b, c). We used GiTools to identify enrichment in gene ontology (GO) terms and pathways (MSigDB Collections), which would be indicative of molecular mechanisms. Cluster 6, including cells re-treated after a 2-day drug holiday, shared many enriched gene signatures with the cells that were treated with erlotinib just for two days, with a notable exception in activation of MAPK cascade and increased regulators of cellular transport and protein secretion such as GAS6, the ligand for AXL receptor tyrosine kinase, a known target for overcoming EGFR-TKI resistance[31] (compare Cluster 6 and D2 cells' Cluster 9 in Fig. 2d and Supplementary Fig. 5b). Clusters 7 and 8 showed resumed expression of the top markers of resistance *CALD1*, *CCDC80*, *TPM1*, *TACSTD2*, and *IGFBP3* (Fig. 2e), as well as of genes with functions in amino acid metabolism ($P < 10^{-5}$) and DNA repair ($P < 10^{-16}$; Fig. 2d and Supplementary Fig. 5c, d). As very similar cell subpopulations emerged after a drug holiday, we concluded that plasticity in drug tolerance is associated with reversibility in DT marker expression.

**Identification of genes associated with EGFR-TKI-resistant cell populations**. In order to test if similar cancer cell populations arise during drug resistance in different NSCLC cells carrying a *EGFR*ex19 mutation, we performed Drop-seq and investigated clusters within untreated samples (D0) and samples treated for 3 days with erlotinib (D3) in PC9 and HCC827 cell lines (Fig. 3a). Certain DT clusters displayed similar enrichments in the same gene sets in both cell lines (e.g., PC9 Clusters 4 and 5 were similar to HCC827 Cluster 6 in Fig. 3b). Overall, 47 of 63 biological processes and pathways that were enriched in HCC827 DT clusters were also enriched in DT clusters of PC9 cells. Similar clusters showed high correlation between RNA and protein level (Supplementary Data 7, 8, 10, and 11), indicating that the common mechanisms may be projected on proteins. These findings imply that while there were different genes at play, the overall molecular mechanism was consistent between specific cell subpopulations. To investigate if common mechanisms may be

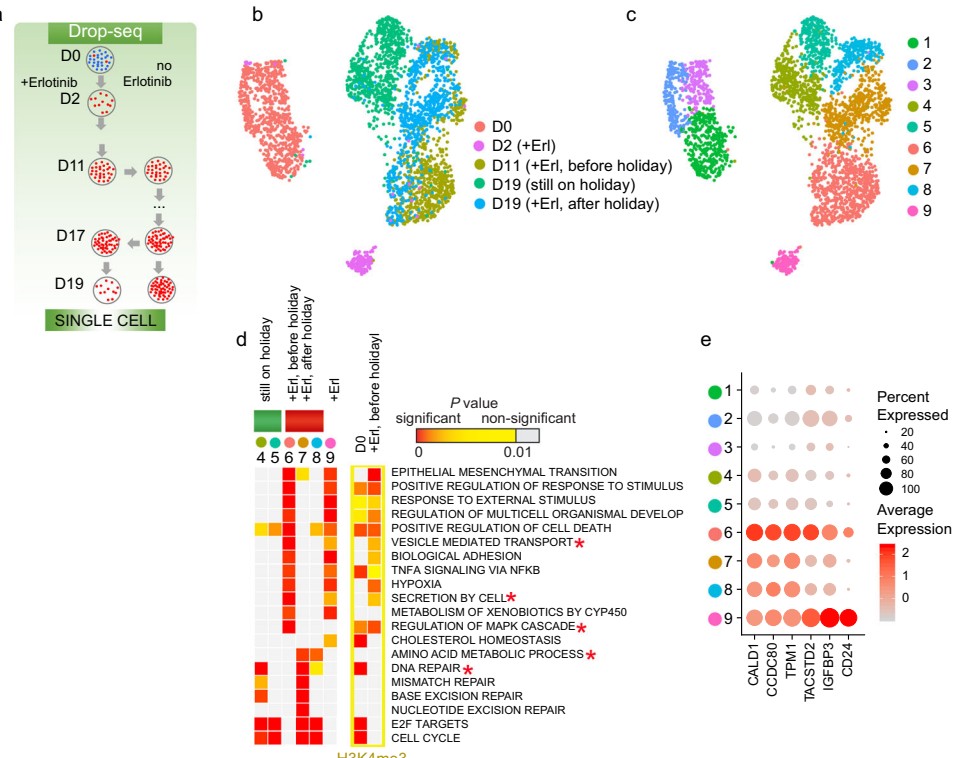

**Fig. 2 Resuming treatment after drug holiday restores tolerant cell populations and their markers. a** Schematics of drug holiday treatment for scRNA-seq. **b** UMAP representation of cells colored by days of treatment. **c** UMAP representation of cells colored by clusters. **d** Enrichment analysis for genes in relation to gene ontology (GO) biological process (BP), hallmark gene sets, or Kyoto encyclopedia of genes and genomes (KEGG) pathways (MSigDB Collections) is shown for top cluster markers (*P* < 0.05). To reveal significance of enrichment of the identified gene set signatures at the level of whole cell population, bulk ChIP-seq data and bulk RNA-seq were used to analyze upregulated genes with increased H3K4me3. Data is shown for large gene sets (>20 genes), with at least 10 markers in one DT cluster, $P < 10^{-5}$, and at least $10^{-3}$ difference with untreated condition. Data represents right tail *P* values, two-sided binomial statistical test, adjusted for multiple testing using Benjamini–Hochberg FDR method. *P* values are delineated in a colored heatmap where color-coding indicates the degree of significance: highest significance (red) to least significance (yellow) and non-significant (gray). Terms that were discussed in the main text are indicated by asterisks. **e** Dot plot of expression for selected markers.

detected across DT clusters generated from different drug treatments, we performed scRNA-seq analysis in two additional models (Fig. 3a and Supplementary Fig. 6a–d). The DT clusters displayed certain cell type-specific gene sets (Supplementary Fig. 6e–h); we observed a more robust interferon (IFN) response in HCC827 cells than in PC9, and pigmentation markers in M14 melanoma cells. However, a majority of highly enriched gene sets were common in all four models: EMT, tissue development, vesicle-mediated transport, and epigenetic regulation (Fig. 3b, c). We emphasize that enrichment of a given process across multiple treatments may involve the activity of different genes (Supplementary Fig. 6e, f).

Previous studies reported the emergence of drug-tolerant clones due to the activation of pathways which bypass normal RTK signaling. While targeting bypass RTKs has been a strategy for developing drug combination treatments, their frequent failure to inhibit cancer cell growth suggests that multiple mechanisms are contributing to resistance[32]. We hypothesized that the multiplicity of resistance mechanisms develops early in treatment. We compiled markers (*P* < 0.05) for each DT state within the set of consecutive samples (Fig. 1d) for their inclusion in MSigDB gene signatures, including the response to various small molecule and genetic perturbations (Supplementary Fig. 7). The earlier DT states (Clusters 4 and 5) resembled various cells exposed to many other types of treatment (Supplementary Fig. 7a). The top signatures, as predicted, were associated with response to EGF and sensitivity to EGFR-TKIs[33,34] (Supplementary Fig. 7a, c). The

transcriptional program initiated by the first EGF pulse[34], including induction of transcription factors (TFs) *EGR1*, *JUNB*, and *FOS*, halted at Cluster 6 (Supplementary Fig. 7c, g and Supplementary Data 1). There was no expression of the *RRM2*, which provides the precursors necessary for DNA synthesis, or the major effector of ATR kinase *CHEK1/CHK1*, and low expression of *CD44*, *ITGA6*, *MKI67*, and *TOP2A*. In contrast, *CDKN1B* and *2B* encoding cyclin-dependent-kinase inhibitors that form complexes with CDK4 or CDK6, were highly induced. The earlier DT states did not feature cell-cycle-related gene sets and E2F targets (Supplementary Fig. 7b, d), which was consistent with a common response to anticancer drugs thoroughly described in previous studies[35,36]. Consistent with activation of pro-survival NF-κB pathway in response to EGFR TKI[37], the NF-*k*B targets *TPM1*, *CALD1*, *FSCN1*, *KRT17*, and *NQO1* were induced, and ~25% of Cluster 4 cells had increased level of TNF-α-related apoptosis inducing ligand *TRAIL/TNFSF10* (Supplementary Fig. 7g and Supplementary Data 1). Activation of MAPK signaling was a prominent signature, consistent with the notion that the activation of PI(3)K–AKT and MAPK pro-survival signaling pathways may induce drug resistance[5], and included *CD24*, *IGFBP3*, *GADD45A*, *TIMP2*, *PSAP*, *DUSP1*, and *PINK1* as markers of all DT states. The regulation of NF-*k*B and MAPK cascades were common to all four models and were observed in the DT clusters enriched in the GO term "regulation of cell death" (Fig. 3b). Thus, anti-apoptotic gene signatures and genes in the NF-*k*B and MAPK pathways are activated in earlier DT states.

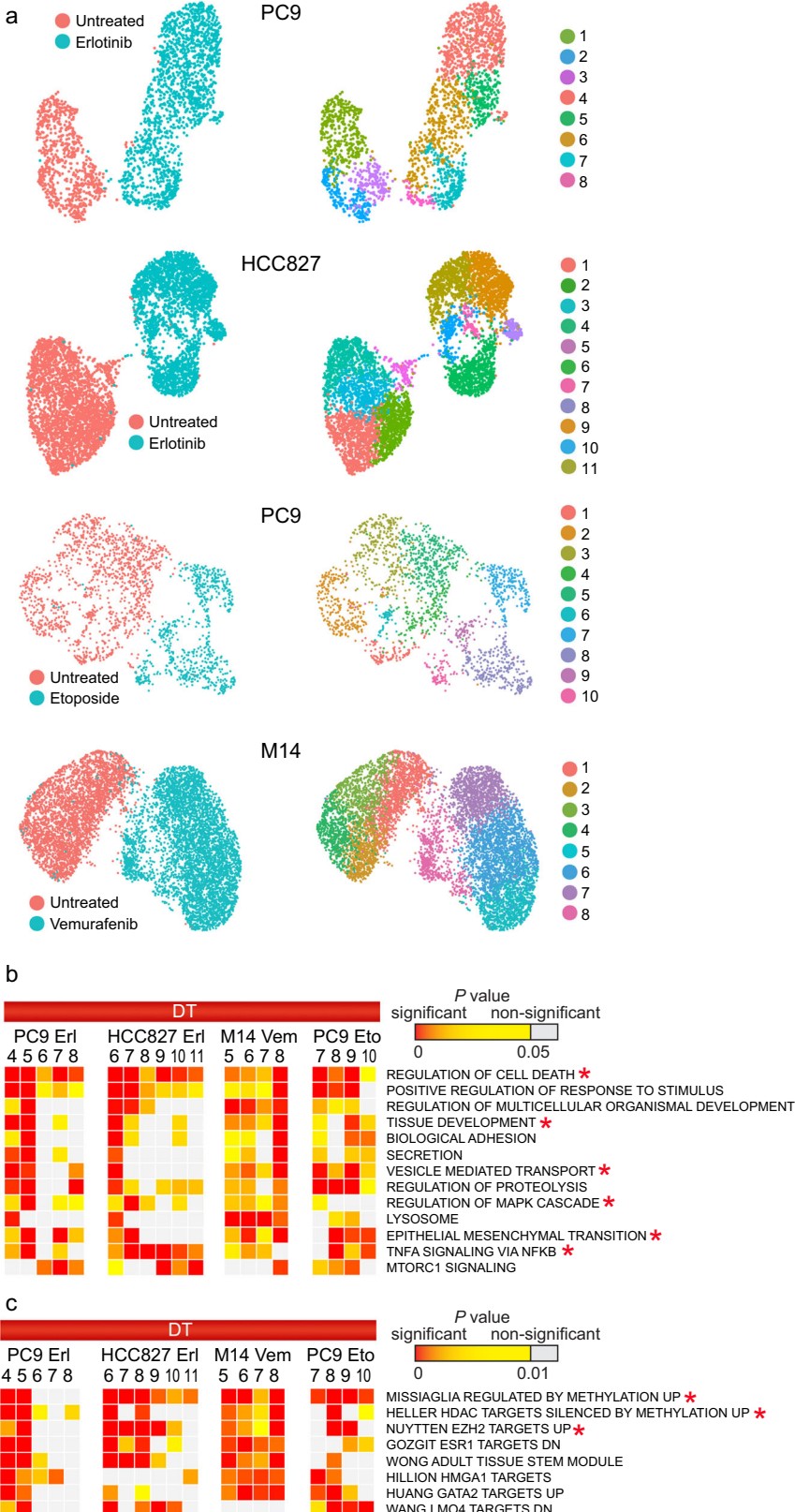

**Fig. 3 Common biological processes can be identified in tolerant subpopulations of distinct cell lines emerging from different treatments. a** UMAP representation of cells in four models of drug tolerance. Cells are colored by days of treatment (left panels) and clusters (right panels). **b** Enrichment analysis for gene relations to GO BPs, KEGG pathways, or hallmark gene sets (MSigDB Collections) is shown for top markers of tolerant clusters ($P < 0.05$). Gene sets appearing highly significant at least in three out of four different treatments and with $P > 10^{-4}$ in any of the clusters of untreated cells are shown. **c** Enrichment analysis for gene sets associated with chemical and genetic perturbations (CGPs). Gene sets that appear highly significant at least in three out of four different treatments are shown. In **b** and **c**, data represents right tail $P$ values, two-sided binomial statistical test, adjusted for multiple testing using Benjamini–Hochberg FDR method. Terms discussed in the text are indicated by asteriscks.

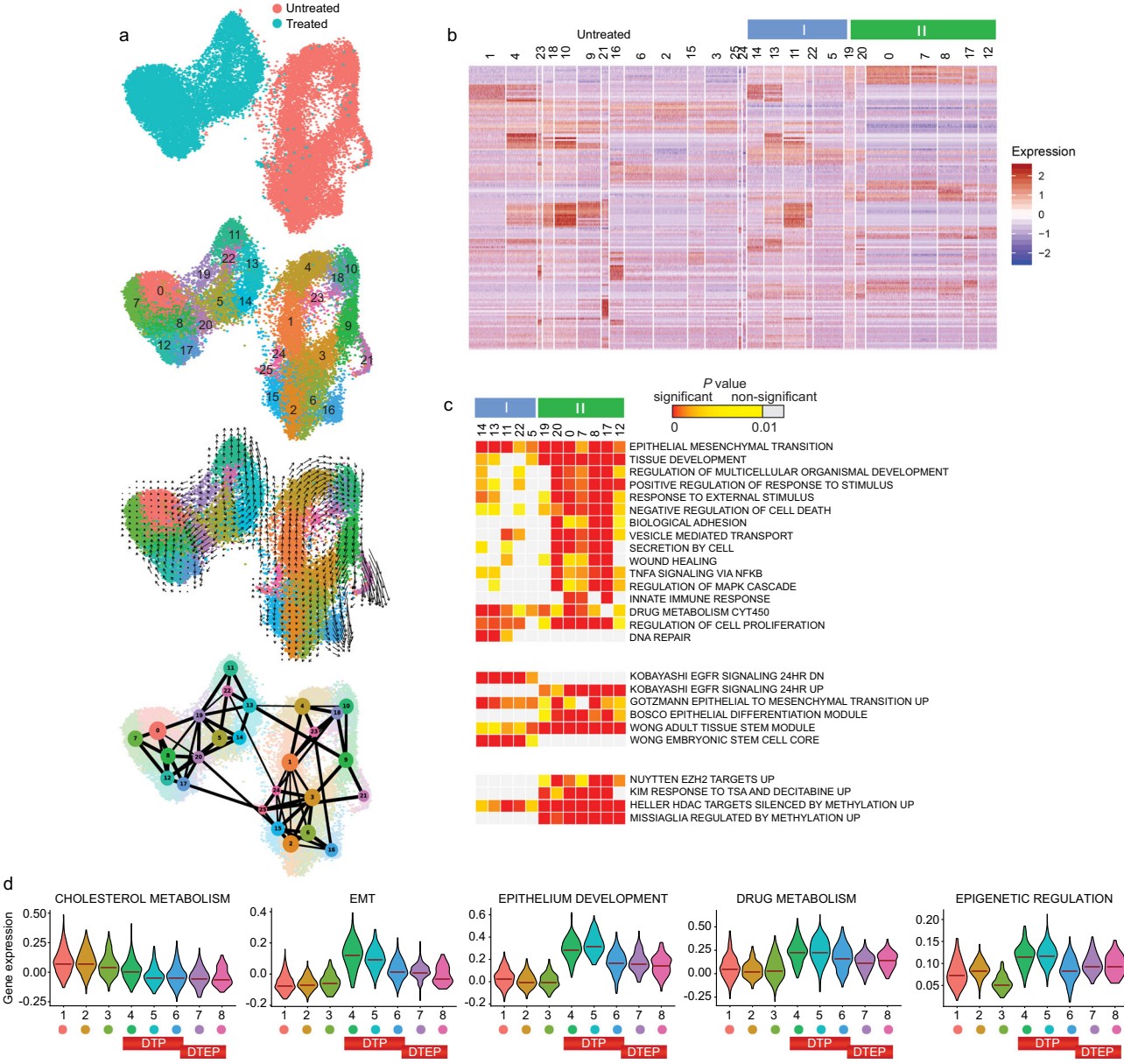

**Fig. 4 Drug tolerant cells are highly heterogenous and represent distinct cell subpopulations and cell states. a** UMAP representation of two PC9 samples, untreated or treated with erlotinib for 3 days, that were analyzed by 10x Genomics scRNA-seq, colored by clusters, with Velocyto projection, and PAGA graph to identify emerging drug-tolerant subpopulations. Arrows reflect the direction in which the nearby cells would travel. PAGA graph is also used to infer trajectory: the size of a circle quantifies the number of cells in the cluster, and the line thickness represents the connectivity strength between clusters. **b** Heatmap of top markers of each cluster. Cluster annotation is as in **a**. **c** Enrichment analysis for top cluster markers ($P < 0.05$). Top, gene relations to hallmark gene sets or GO BP terms; middle, gene sets associated with CGP datasets; and bottom, epigenetic signatures (MSigDB Collections). Two subpopulations of tolerant cells, I and II, are delineated by blue and green boxes. GO gene sets overlapping with at least 10 markers in one DT cluster, $P < 10^{-6}$, were included. Most presented CGPs are representative of several similar experiments, where terms with more than 18 markers and $P < 10^{-8}$ at least in one DT cluster, but $P > 10^{-4}$ in any of the cluster of untreated cells were included. Data represents right tail $P$ values, two-sided binomial statistical test, adjusted for multiple testing using Benjamini–Hochberg FDR method. **d** Violin plots of expression level for gene sets corresponding to top biological processes, hallmarks, and epigenetic perturbations (MSigDB Collections) in cells at different states of transition to tolerance (from Fig. 1d). The median of the data is shown by the horizontal line.

We suggested that the different processes and pathways contributing to drug tolerance are distributed between distinct subpopulations rather than co-existing in individual cells. We captured 23,415 untreated cells and cells treated with erlotinib for 3 days (D3) using 10x Genomics, which enabled the Seurat clustering algorithm to distinguish twelve DT clusters within the Day 3 DT state (Fig. 4a, b and Supplementary Data 3). To avoid false conclusions about heterogeneity of the cell population, the clustering results were accessed with respect to silhouette widths[23] (Supplementary Fig. 8). Velocyto showed directionality within two large subpopulations of DT clusters, and Partition-based Graph Abstraction (PAGA) graph aided in interpretation of connections. The projected directionality was consistent with the order of cell cycle progression (Supplementary Fig. 9a). The two subpopulations were clearly distinguishable by the pattern of enriched processes and gene sets across cell clusters (Fig. 4c).

GiTools analysis found a clear difference between the subpopulations I and II in response to stimulus and activation of the NF-κB and MAPK pathways, and in drug metabolism, epigenetic regulation and putative transcription factors involved (Fig. 4c and Supplementary Fig. 9a–d). The subpopulation II was more responsive to epigenetic inhibition than subpopulation I, suggesting that the appearance of DT markers depends on activity of epigenetic enzymes. Signaling pathways leading to EMT, including transforming growth factor-β (TGF-β) family proteins TGFB2, INHBA, and INHBB, plasminogen activator protein inhibitor SERPINE1, and WNT/β-catenin pathways, represented an overarching category of gene sets enriched in the DT clusters (Fig. 4c and individual markers in Supplementary Fig. 9e and Supplementary Data 36 and 37). Using Monocle[38], we identified several sets of co-regulated genes, referred to as modules, by comparing clusters of treated and untreated cells (Supplementary Fig. 9f, g, and Supplementary Data 12). DT cells lacked expression of Modules 1 and 2 which are related to cell cycle regulation and cholesterol metabolism ($P < 10^{-15}$), but expressed three EMT modules: (1) INHBA$^{high}$ module ($P = 3.9 \times 10^{-11}$); (2) MET$^{high}$ module ($P < 10^{-15}$); and (3) TPM1$^{high}$/TPM4$^{high}$ module ($P = 3.4 \times 10^{-8}$). Notably, the MET$^{high}$ module was enriched for the focal adhesion pathway ($P = 6.1 \times 10^{-9}$; Supplementary Fig. 9h). The TPM$^{high}$ module was similarly characterized by high expression of CDH2, COL1A1, and CALD1, and genes related to apical junction complex function. The upregulation at the transcript level in cell adhesion and apical junction gene is likely to result in functional change, as these gene sets show the greatest correlation between RNA and protein level[29]. Thus, in response to EGFR TKI, cells lost expression of genes involved in cholesterol metabolism. In tolerant cells, we found a high enrichment in EMT, in tissue development including epithelium development, drug metabolism, and epigenetic regulation (Fig. 4d and Table 1). There was high difference in enrichment of these four processes across distinct subpopulations, suggesting that different mechanisms may push cells towards drug tolerance. These processes have not yet been extensively linked to drug resistance. However, we observed them using scRNA-seq in four drug tolerance models (Fig. 3b, c).

**Selecting inhibitors targeting DT markers.** We asked if inhibiting an individual top marker of a DT state was sufficient to suppress cell growth under treatment with erlotinib. We found that dose response to erlotinib compared to DMSO in cells transduced with TACSTD2- or SERPINE1-specific inhibitory RNAs showed no significant difference from cells transduced with negative control RNAs, despite efficient knockdown and knockout (Supplementary Fig. 10a–d). Similarly, there was no dramatic difference between the experimental and negative control RNAs in 21 day-long cell survival assays (Supplementary Fig. 10e). Additionally, transduction of either PC9 or HCC827 cells with a combination of TACSTD2 and SERPINE1 shRNA lentiviruses provided no more growth inhibition under erlotinib than the combination of control shRNAs (Supplementary Fig. 10f, g). Thus, the inhibition of individual top markers was not sufficient for impairing survival with erlotinib.

Thus, we argued that downregulating multiple markers would sensitize cells to the drug. We asked if multiple markers of DT states could be targeted with a small molecule. Such molecule would potentially revert the tolerance and inhibit the growth of DT cells when combined with erlotinib. We considered that DT markers may be parts of upregulated networks that can be attacked effectively by drugs which downregulate those networks. The Library of Integrated Network-based Cellular Signatures (LINCS) that catalogs gene expression responses to 268 small-molecule inhibitors in 15 cancer cell lines was analyzed as described in the Methods section. The list of markers of each DT state was compared to the LINCS drug response signatures (Fig. 5a). We found highly significant downregulation of markers of Clusters 4, 5, 7, and 8 among LINCS signatures (Fig. 5b and Supplementary Data 13). Crizotinib and celastrol were the top two inhibitors identified across different clusters. Many drugs showed highly significant values for a single cluster, including a reversible pan-PI3K/mTOR inhibitor GSK1059615 that inhibits the phosphorylation of Akt at S473. Identification of GSK1059615 is relevant as erlotinib sensitivity has been previously correlated with failure of EGFR to couple to downstream survival signals. Using bulk RNA-seq data we were able to identify celastrol and GSK1059615. But in bulk RNA-seq data the full list of identified targeting drugs was much shorter than those generated using scRNA-seq cluster analysis and did not include crizotinib (Supplementary Data 13–16). The LINCS signatures downregulated by crizotinib, celastrol and GSK1059615 were preferentially expressed in DT clusters but not in untreated cells (Fig. 5c and Supplementary Fig. 11a). GiTools enrichment analysis showed that these drugs effectively target the processes and TFs pertinent to the DT states (Fig. 5d, e), including top transcriptional signatures induced in DT states (Supplementary Fig. 7d, f), targets of lymphoid enhancer-binding factor 1 (LEF1/TCF7L1) which favors EMT and ROS-responsive nuclear factor of activated T cells (NFAT). As we have mapped DT markers onto known biological pathways (Fig. 4d), we next asked whether they are common activated signatures in the LINCS drug response database. Activation of EGFR and NF-κB signaling, EMT, epithelium development, and/or epigenetic signatures was a common drug response (Supplementary Data 17). Thus, LINCS analyses inferred drugs that may be synergistic when combined with erlotinib.

Addition of either crizotinib or GSK1059615 decreased viability of PC9 DTPs as well as DTEPs (D3 cells and D11 cells in Fig. 5f, g, and Supplementary Fig. 11b–d). The decrease in cell viability in PC9 DTPs was at least partially due to an increased apoptosis (Supplementary Fig. 11e). The sensitivity to crizotinib or celastrol was comparable to sensitivity to AEW541 (Fig. 5g), the IGF-1R inhibitor known to be effective in growth inhibition of PC9 DTEPs[15]. Consistent with the LINCS identification of drugs targeting specific clusters in both PC9 and HCC827 cells (Supplementary Data 15 and 16), their combination with erlotinib was effective in cell survival assays (Fig. 5f, g and Supplementary Fig. 11b, e). Thus, we have validated crizotinib, celastrol and GSK1059615 as combination agents with EGFR TKI. LINCS analysis identified celastrol as a common top drug and crizotinib as a less significant drug, downregulating DT markers in two other drug tolerance models (Supplementary Data 18 and 19). Celastrol and crizotinib were confirmed in survival assays, in contrast to GSK1059615, which was not potent in decreasing cell survival in PC9 Eto or M14 Vem (Supplementary Fig. 11f, g).

As the cell survival assays revealed residual double tolerant cells, we proposed that they may arise from the DT clusters that were not targeted by the predicted drug. Therefore, we again employed scRNA-seq (Supplementary Fig. 12), to examine which erlotinib-resistant cells died from the combination treatment (Fig. 6a). We detected two groups of clusters: tolerant to Erl but sensitive to Criz, Criz-S, and tolerant to both Erl and Criz, Criz-T (Fig. 6b, Supplementary Fig. 13, and Supplementary Data 20). LINCS analysis showed that crizotinib specifically downregulated Criz-S cluster markers but not Criz-T cluster markers (Supplementary Data 21). This result is significant because it shows that one can design a drug that selectively targets predicted cell subpopulations. Furthermore, Criz-T clusters were not

**Table 1 Hallmarks of erlotinib resistance in EGFR-mutant cell lines.**

| | |
|---|---|
| Cholesterol metabolism | • Enzymes of almost the whole pathway, from acetoacetyl coenzyme A transferase (ACAT2) and CYP51A1 all of the way down to cholesterol, were expressed at gene level relatively lower in DT cells, together with the low-density lipoprotein receptor (LDLR) responsible for cholesterol uptake, STARD4, involved in intracellular cholesterol transport, the sterol-sensing protein INSIG1, and sterol response element-binding proteins (SREBPs) SREBF-1 and 2, while the ATP-binding cassette A1 (ABCA1) involved in lipid efflux from cells, was upregulated.<br>• The SREBP targets *FASN, FDFT1, FDPS, HMGCS1, HSD17B7, IDI1, INSIG1, LDLR, PCSK9, RDH11, SQLE, STARD4*, which are directly involved in cholesterol biosynthesis[128], were downregulated in all DT clusters.<br>• Biosynthesis of unsaturated fatty acids that involves regulation through PPAR-γ signaling pathway (*ACSL1, ACSL3, ACSL4, FADS1, FADS2, ME1, SCD*) is likely to be impaired in DT clusters ($P = 5.4 \times 10^{-12}$). Supplementary Figs. 7e and 9d |
| EMT | • Among the transcription factors that regulate EMT[56], *SOX4, SLUG/SNAI2*, and *GATA6* were highly increased in DT cells. Analysis for enrichment of TF-binding sites identified lymphoid enhancer-binding factor 1 (LEF1/TCF7L1) that favors EMT and is downstream of WNT/β-catenin.<br>• The *TGFB2* marker expression correlated with genes involved in protein secretion and increased activity of promoters occupied by SMAD2 or SMAD3[129] (73 promoters in Cluster 4 and 68 promoters in Cluster 5, Supplementary Fig. 7c), indicating SMAD pathway activation[130,131]. There was an increase in the transactivator for TGF-β-dependent transcription *CITED2*.<br>• *IGFBP3*, the main carrier protein for insulin-like growth factors (IGF), along with the homologous protein *IGFBP5*, were increased.<br>• Features of senescence associated secretory phenotype (SASP)[132] included growth arrest and secretion of extracellular matrix proteins such as fibronectin *FN1* and the CCN family of matricellular ligand *CYR61/CCN1*, which binds to integrins and has been linked to chemotherapy resistance, and proteases such as serpin E1 *SERPINE1/PAI1*[25]. SASP initiated stimulator of interferon-γ (IFN-γ) genes (STING) response, enhancing the production of chemokines and type I interferons.<br>• Many markers functionally belonged to vesicle-mediated transport.<br>• Genes encoding cytokines and associated proteins such as *SPARC* were highly increased. SPARC regulates cell growth through interactions with the extracellular matrix oxidized low density lipoprotein (oxLDL) receptor 1 OLR1. Its association with oxLDL induces the activation of NF-κB.<br>• Focal adhesion pathway (*FN1, COL5A1, MYL12A, PDGFC, TNC, IGF1R, COL4A4, SHC3, PXN, CAV1, CAV2, LAMC1, ITGA4, MET, FYN*) or apical junction complex (*ACTB, ACTN1, ACTN4, ACTG2, MYL9, RSU1, NEXN, LIMA1, ZYX, VCL, CNN2, MYH9*) were characteristics of MET^high and TPM^high modules.<br>Figs. 2d, 3b, 4c, 5d, e, 6c, f, and 7b, d and Supplementary Fig. 7b–f, 9c, d, h, and 14f |
| Epithelium development | • Enrichment of GO terms related to tissue development, multicellular organismal development and CGP gene sets related to lung epithelium differentiation[133].<br>• Upregulation of TFs such as *THBS1, FOXJ1/HFH4, JUN, KLF9* and *NFIB* that influence epithelial cell growth and differentiation[134].<br>• A shift towards the GATA6-high program and repression of putative markers of distal airway epithelium *STEAP1, GPR87*, vascular and ECM remodeling genes *VEGFA* and *PLAU*, and cytokeratins *KRT6A* and *KRT6B*[135]. Figs. 2d, 3b, 4c, 6c, d, and 7b, d and Supplementary Fig. 7b, c. |
| Drug metabolism | • TF-binding site analysis predicted activation of nuclear factor of activated T cells (NFAT), which is indicative of the imbalance in superoxide/hydrogen peroxide.<br>• Upregulation of the ROS genes *GPX4* and *PDLIM1* across all DT clusters.<br>• Upregulation of targets downstream of PI3K/AKT/mTOR, *SQSTM1*, and *AP2M1*, which may trigger synthesis of detoxifying enzymes that prevent oxidative stress. SLC3A2 in complex with SLC7A5 imports essential amino acids and promotes mTORC1 activity[136]. SLC3A2 and SLC7A5 were both upregulated, which may be important in managing erlotinib-induced ROS, as it was accompanied by an increase in the high-affinity glutamine transporter *SLC1A5* and *SLC7A11*, whose products form the cystine transporter complex with SLC3A2, and are required for glutathione synthesis.<br>• Enrichment for drug metabolism through cytochrome P450, represented by *GSTK1, GSTM3, GSTM4, MGST3, ALDH1A3, ALDH3A1, ALDH3B1*, and *CYP1B1*. Figs. 2d, 4c, 6c, e, and 7b, d, and Supplementary Fig. 7b, d, f |
| Epigenetic regulation | • Enrichment of upregulated EZH2 targets (e.g., $P < 10^{-16}$, 124 genes[137]).<br>• Enrichment of upregulated targets of class I and II HDACs (e.g., $P < 10^{-16}$, 71 genes[138]).<br>• Enrichment of gene sets associated with inhibition of DNA methylation (e.g., $P < 10^{-16}$, 37 genes[139]). The P values are provided from data in Fig. 4c. See also Figs. 3c, and 6d, and Supplementary Fig. 14e, l. |

Top genes and enriched processes and gene sets are shown. See Supplementary Data 1, 3, 12, 20, and 30–55 for individual markers.

independent clusters but represented a fraction of Erl DT clusters. This suggested that combined treatment did not generate a significantly differential transcriptional response that would result in novel surviving cell subpopulations. In fact, Criz-T markers significantly overlapped with terms from MSigDB Collections which were identified above for erlotinib DT markers, including the EGFR-TKI resistance signature[33] (Fig. 6c, d, e.g., *ALDH1A3* and *TACSTD2* in Supplementary Fig. 13e) and were highly expressed in erlotinib DTP states (Supplementary Fig. 13g). We were able to detect upregulation of markers that distinguish Criz-T clusters from Criz-S

clusters (*CTSA, GSTK1, PDLIM1, PSAP, TMEM59, CYP1B1*, and *FAM134B*), but not *OLR1* and *TGFB2* that fail to distinguish these clusters, in bulk RNA analysis (Fig. 6g).

The LINCS analysis identified celastrol as an inhibitor of Criz-T markers (Supplementary Data 21). smRNA-FISH showed that while cells that were not sensitive to the addition of crizotinib had high number of *TACSTD2* transcripts, such cells disappeared after trio combination treatment (Fig. 6h, i). An 11-day-treatment dramatically decreased cell viability (Fig. 6j), and addition of crizotinib and celastrol 48 h after the start of erlotinib treatment resulted in the greatest reduction of cell numbers (Fig. 6k).

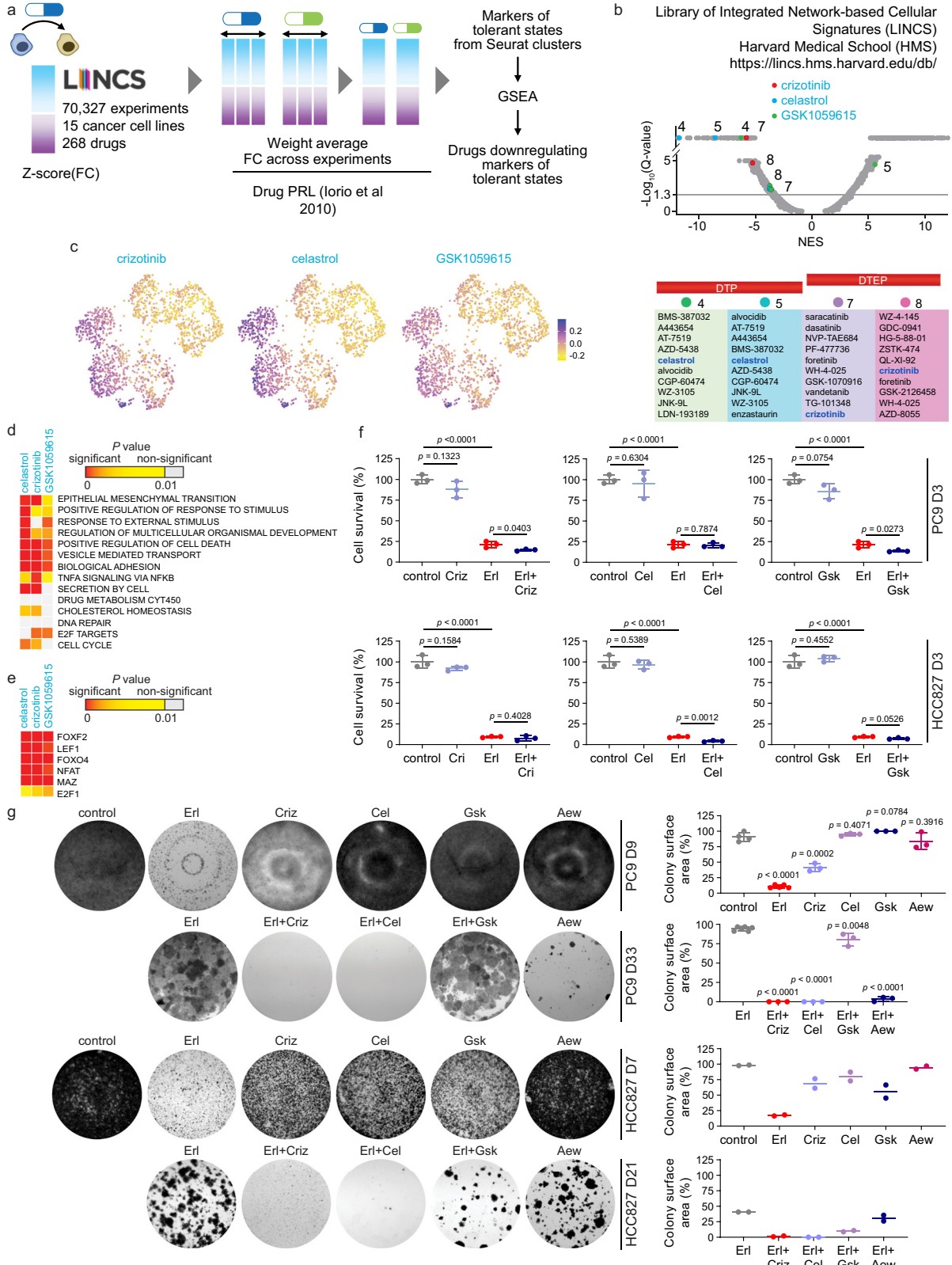

Therefore, celastrol specifically targets Criz-T cells and, with the proper attention to underlying gene expression patterns associated with cell heterogeneity, drug combinations may be effective in targeting processes in cells resistant to pairings.

Crizotinib is a first-generation ALK inhibitor which is clinically utilized as a mesenchymal epithelial transition (MET) RTK

inhibitor. Most EGFR-TKI acquired resistant tumors display MET activation, which occurs via increased transcription and protein expression of *MET*. Our identification of crizotinib was highly relevant because previous compound screenings showed that the EGFR/MET inhibitor combination was effective for eliminating resistant cells[39]. HCC827 cells have amplification of

**Fig. 5 Using LINCS analysis to identify effective drug combinations. a** Identifying candidate molecules for drug combinations using LINCS. The gene expression values were calculated from experiments in LINCS database and used to select drugs that would downregulate the genes identified as markers in this study. Description is provided in the Methods section. **b** Normalized enrichment scores (NES) for LINCS drugs generated using GSEA. NES corresponds to weighted Kolmogorov–Smirnov-like statistic, FDR was computed by comparing the tails of the observed and null distributions. Relative positioning of the top drugs, crizotinib, celastrol, and GSK1059615, which would significantly downregulate markers of states 4, 5, 7, and 8, are shown on the NES plot. *Y*-axis tick values depict the levels of significance, with $-\text{Log}10(Q\text{-values})$: <1.3 – non-significant, >1.3 but <5 – significant, and >5 – highly significant. Top 10 drugs downregulating tolerant states are listed at the bottom. **c** Feature plot showing cells colored by score of responsiveness to a drug. The score was calculated by Seurat for the DT markers that would be responsive to the drug. The UMAP is from Fig. 1. **d**, **e** Enrichment of gene relations to the terms from MSigDB Collections described for tolerant states among the markers decreased by the drugs. Data represents right tail *P* values, two-sided binomial statistical test, adjusted for multiple testing using Benjamini–Hochberg FDR method. **e** shows enrichment of transcription factor targets (TFTs). Each column in **d** and **e** represents PC9 DT markers that LINCS analysis predicts to be downregulated by the drug. **f** Survival assays of PC9 and HCC827 cells treated for 3 days with crizotinib (Criz, 1 μM), celastrol (Cel, 2 μM), and GSK1059615 (Gsk, 1 μM) alone or in combination with erlotinib (1 μM for PC9 and 30 nM for HCC827). Data represents mean ± SD (*n* = 3). **g** Colony formation assays of PC9 and HCC827 cells treated for indicated number of days with drugs at the concentrations as in **f**. The IGF-1R inhibitor AEW541 was used at 1 μM as a control. Representative crystal violet staining of two independent experiments is shown. Plate colony surface area is shown as mean ± SD for *n* = at least three replicate PC9 wells and as mean values for *n* = 2 replicate HCC827 wells. In **f** and **g**, two-tailed *P* values were determined by unpaired *t* test relative to the DMSO control or Erl via GraphPad Prism 7.

the *MET* locus, which likely accounts for their increased survival compared to PC9 cells treated with an equal concentration of crizotinib (Fig. 5f, g). Criz-S clusters displayed increased expression of *MET* compared to Criz-T clusters (Supplementary Fig. 13e). The EMT- and TGF-β receptor signaling-related gene signatures appeared only in the Criz-S clusters (Fig. 6c–e and Supplementary Fig. 13e). This data is consistent with our identification of the MET^high module (Supplementary Fig. 9f–h) and suggests that MET overexpression is driving resistance in specific cell subpopulations. Instead, the Criz-T markers highly expressed the epithelial G protein-coupled receptor *GPRC5A* and were enriched in gene sets related to epithelium development, drug metabolism, lysosome, and epigenetic signatures, suggesting that Criz-T clusters correspond to the subpopulation II of Erl-treated cells (Fig. 4c). It is possible that celastrol sensitized Criz-T clusters, because it is known to be a potent inhibitor of the NF-κB activation and Criz-T Cluster 8 markers were enriched in binding sites of RELA sub-unit of NF-κB (Fig. 6f), as well as genes involved in drug metabolism through cytochrome P450, which are also Criz-T markers (Fig. 6c, e).

Since cells grown in culture lack many in vivo interactions, we modeled PC9 cell response to EGFR TKI in a xenograft study (Supplementary Fig. 14a, b). Osimertinib decreased tumor volume and the EGFR TKI/crizotinib combination further decreased tumor size, which was consistent with previous studies[40,41]. scRNA-seq analysis of tumor tissues showed that osimertinib treatment produced several novel cell populations compared to vehicle-treated animals (Supplementary Fig. 14c), with many of the top markers (*GPRC5A, SOX4, FOS, JUN, IGFBP3,* and *ALDH3A1*), gene signatures (Fig. 7a, b and Supplementary Fig. 14d–h) and LINCS small molecules (Supplementary Data 22) also observed in the PC9 cell culture model. Specific xenograft cell populations showed induction of the marker of alternative cell program claudin−4 (*CLDN4*)[42], *CLDN7*, and mucins such as *MUC16* (Supplementary Fig. 14h), which was consistent with the high enrichment in terms related to tissue development. Combination of osimertinib and crizotinib eliminated cells in specific clusters, including a likely proliferative osimertinib-resistant Cluster 8 cell population whose clustering depended on repressing cell cycle genes (Supplementary Fig. 14i). Thus, combination treatment in the xenograft model specified Criz-S and Criz-T cluster groups (Fig. 7c). GiTools analysis identified gene signatures similar to the cells grown in culture, although the xenograft Criz-T clusters developed increased expression of genes involved in antigen presentation (Fig. 7d and Supplementary Fig. 14j–m). No new subpopulations

appeared after treatment with the second drug, and we concluded that combination treatment may reduce heterogeneity of the cell population by downregulating specific survival mechanisms.

**Tolerance markers in conventional patient datasets and in scRNA-seq data from fresh patient tumors.** We then tested if DT cluster markers were more broadly clinically relevant by examining their expression level in patient tumor tissues. We used a bulk microarray gene expression dataset of 127 NSCLC patients with EGFR mutations for which survival data was available[43,44]. We applied sample-level enrichment analysis (SLEA)[45] to calculate *Z*-scores for the DT markers. High *Z*-scores revealed that DT marker expression was preferentially increased in many patients (Fig. 8a). Patients with the increased expression of individual DT state markers or all combined DT markers displayed a decrease in overall survival (Fig. 8b and Supplementary Fig. 15a). The difference in survival was not only due to cell proliferation genes, as it was also observed for the DTP Cluster 6 (*P* = 0.0034) which lacks cell cycle signatures (Supplementary Fig. 7b–d). The markers of intrinsic cell populations differed in statistical power for predicting survival; the markers of the osimertinib-resistant proliferating Cluster 8 cell population showed especially high *Z*-scores and decreased survival (Supplementary Fig. 16a, b). Vemurafenib DT cluster markers were imperfect in distinguishing patient survival of melanoma, a cancer with much longer survival (Supplementary Fig. 16c–f).

The significant association of DT markers with patient survival may be a reflection of pre-existing and acquired drug resistance and indicates the usefulness of scRNA-seq analysis in identifying cancer cell populations in the NSCLC tumor tissue. Thus, we sought to ascertain the clinical value of the genes which distinguish cancer cells from other cells. A tissue processing and analytical pipeline has been developed for single-cell analysis of NSCLC tumors to identify: (1) tumor-specific cell populations; (2) markers of epithelial cancer cell populations; and (3) drugs that specifically target the cancer cell populations and not normal cells.

We subjected three NSCLC tumors resected from different patients to Drop-seq analysis. One carried the *EGFR*ex19 mutation, another had the *KRAS*G12C mutation, and the third tumor contained multiple oncogenic driver mutations (Supplementary Data 23). Cell clusters were annotated using gene expression data of known cell types (Supplementary Data 24). Overall, we captured 4328 cells, which belonged to multiple clusters (Supplementary Fig. 15b, c). Based on the top differentially expressed genes (Supplementary Data 25), we were able to assign

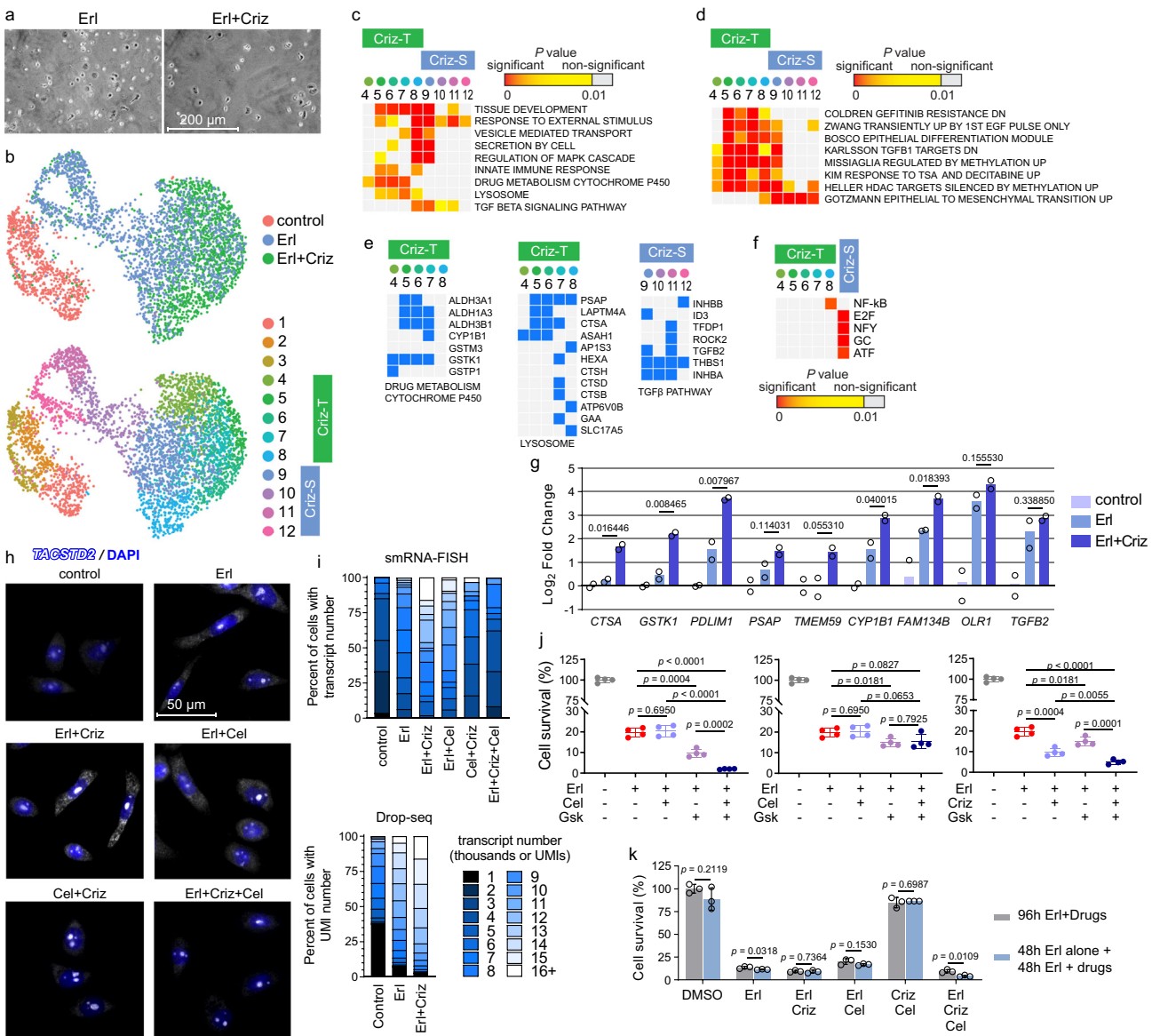

**Fig. 6 Crizotinib-tolerant cells represent a subpopulation of erlotinib-tolerant cells. a, b** PC9 cells were treated for 3 days with 2 μM erlotinib (Erl), with 1 μM erlotinib and 1 μM crizotinib (Erl + Criz) or left untreated (control). **a** Representative microscopic images of the cells. The experiment was repeated six times independently with similar results. **b** UMAP representation of PC9 cells colored by treatment and by crizotinib-tolerant (Criz-T) and crizotinib-sensitive (Criz-S) clusters. $n = 2$ biological replicates. **c** Enrichment analysis for gene relations to GO BP and KEGG pathways terms is shown for top DT cluster markers ($P < 0.05$). **d** Enrichment analysis for gene sets associated with CGPs. Gene sets with $P < 10^{-7}$ and >11 markers in a DT cluster but $P > 10^{-4}$ in any untreated cluster are shown. **e** Criz-T and Criz-S markers in top enriched terms. **f** Occurrences of TF-binding sites (TFBSs) from TRANSFAC database in promoters of Criz-T cluster markers and combined Criz-S markers. Enriched TFBSs with corrected $P < 10^{-9}$ are shown in the heatmap. In **c**, **d** and **f**, data represents right tail $P$ values, two-sided binomial statistical test, adjusted for multiple testing using Benjamini–Hochberg FDR method. **g** Gene expression changes in the levels of Criz-S and Criz-T markers using bulk RNA samples. The RT-qPCR data was normalized to *POLR2B* level and presented as $\text{Log}_2$ fold change relative to DMSO-treated control cells, mean for $n = 2$ biological replicates. **h** smRNA-FISH reveals high *TACSTD2* expression as a characteristic of Erl + Criz-tolerant cells, which is abrogated by celastrol. The experiment was repeated two times independently with similar results. **i** Quantitation of fluorescent microscopy images (at the top) and Drop-seq data (at the bottom) showing *TACSTD2* transcript abundance. Scaled average expression across cells is shown by the color of the bar, with a darker blue representing a low expression, and white representing a high expression. **j** Survival assays of PC9 cells treated for 11 days with the drugs as in Fig. 5f. Data represents mean ± SD for $n = 4$ replicate wells. **k** Survival assays of PC9 cells subjected to successive drug treatment. Data represents mean ± SD ($n = 3$). In **j** and **k**, two-tailed $P$ values were determined by unpaired $t$ test relative to the simultaneous treatment via GraphPad Prism 7.

identities of >99% cells to 16 different cell types. We included previously reported data from three donors[46] in our analysis to compare tumor cells to lung tissue without disease. Non-hematopoietic cells were subset and further clustered, leading to identification of nonmalignant and cancer epithelial cells (Fig. 8c–e and Supplementary Data 26). Consistent with previous

reports in other tumors (reviewed by Suva and co-authors[47]), cancer cells were patient-specific (Fig. 8e). Following the identification of cancer epithelial cells, we ascertained their similarity to other tissues in order to identify the cell of origin. In comparison to ciliated, Club/Clara cells, pulmonary alveolar type 1 (AT1) and pulmonary alveolar type 2 (AT2) (Fig. 8d), the

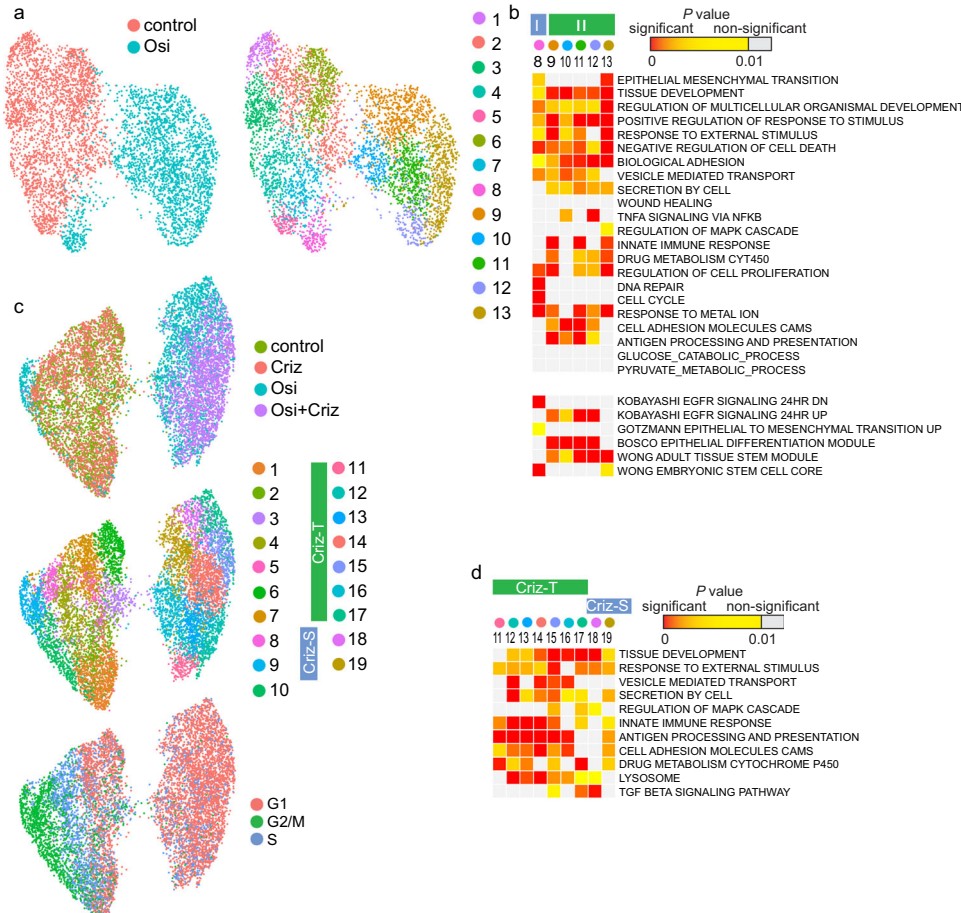

**Fig. 7 Crizotinib inhibits the emergence of a defined subset of osimertinib-tolerant cell populations in vivo. a** UMAP representation of tumor cells from mice treated with osimertinib for 3 days and control mice treated with vehicle, colored by clusters (on the right). **b** Enrichment analysis for top cluster markers ($P < 0.05$). Data are presented for the terms in Fig. 4c and any additional gene set with $P < 3×10^{-6}$ in most of the clusters of either osimertinib-treated or control tumors. Two subpopulations of tolerant cells, I and II, are delineated by blue and green boxes. **c** UMAP representation of tumor cells from mice treated with the combination of crizotinib and osimertinib, each drug alone and vehicle control. All mice were treated for 3 days. Treatment replicates are colored by the drug (at the top), by clusters (in the middle), and the cell cycle phase positioning of each cell (at the bottom). **d** Enrichment analysis for markers of DT clusters, which are either crizotinib-tolerant (Criz-T) or crizotinib-sensitive (Criz-S). Data is presented for the terms in Fig. 6c and any additional gene set with at least $10^{-6}$ difference in $P$ value in majority of the Criz-T clusters from Criz-S clusters and control tumors. In **b** and **d**, data represents right tail $P$ values, two-sided binomial statistical test, adjusted for multiple testing using Benjamini–Hochberg FDR method; gene relations to GO BP terms, KEGG pathways and hallmark gene sets (MSigDB collections) are shown for top markers ($P < 0.05$).

*EGFR*ex19 cancer cell cluster was characterized by high expression of *NPC2* and surfactant genes *SFTPB*, *SFTPC*, and *SFTPD* (Supplementary Data 26 and 27). These genes represented the markers of AT2 cells, which normally form the lining of the alveoli. Other distinguishing markers were cathepsin D (*CTSD*) and *APOD*. This data was consistent with the notion that AT2 cells are cell-of-origin for *EGFR*-mutant NSCLC[48]. Unlike the AT2 cells, the cancer cell cluster did not express the SFTPA gene. Fibroblasts in the *EGFR*ex19 patient had particularly distinctive markers in comparison to fibroblasts from two other patients. Their markers, a member of small leucine-rich proteoglycan (SLRP) family and Serpin F1, indicate that these fibroblasts may relate to AT2-derived cancer cells through the activation of specific processes such as WNT signaling pathway[49].

scRNA-seq data on tumor cells has potential clinical importance for identifying small molecules to target cancer cell clusters. Applying LINCS to *EGFR*ex19 or *KRAS*G12C markers, we obtained a number of highly significant drugs in the GSEA ranking (Supplementary Data 28 and Fig. 8f). The top 10 drugs

for the *EGFR*ex19 included various CDK inhibitors and the AKT inhibitor A443654. These findings emphasize the utility of single-cell gene expression data in identification of drugs specific to the biology of activated pathways. We also analyzed the tumor cell data for enrichment in GO terms and CGPs for comparison with our findings in cell lines. *EGFR*ex19 and *KRAS*G12C cancer cell clusters displayed differential enrichment in lung cancer signatures, proliferation signatures, epithelium development, and TGF-β signaling through SMAD2 and SMAD3, when compared to other epithelial clusters (Fig. 8g, h). From the top ten drugs (NES < −4.2, $P < 0.005$) for the *EGFR*ex19 tumor, six molecules overlapped with top 10 LINCS drugs predicted for DT PC9 clusters (Fig. 5b). One possibility is that six drugs are common false positives in the dataset. However, for the *KRAS*G12C tumor, LINCS analysis returned different drugs (NES < −3.6, $P < 0.04$), inhibitors of HSP70, GSK3β, and KRAS signal transducers BRAF and RAF1. The *EGFR*ex19 cancer cell clusters showed exclusive expression of the markers targeted by A443654 and AT-7519 (Fig. 8i, j and Supplementary Fig. 15d, e), but not of the erlotinib-induced genes targeted by the same drugs (Supplementary

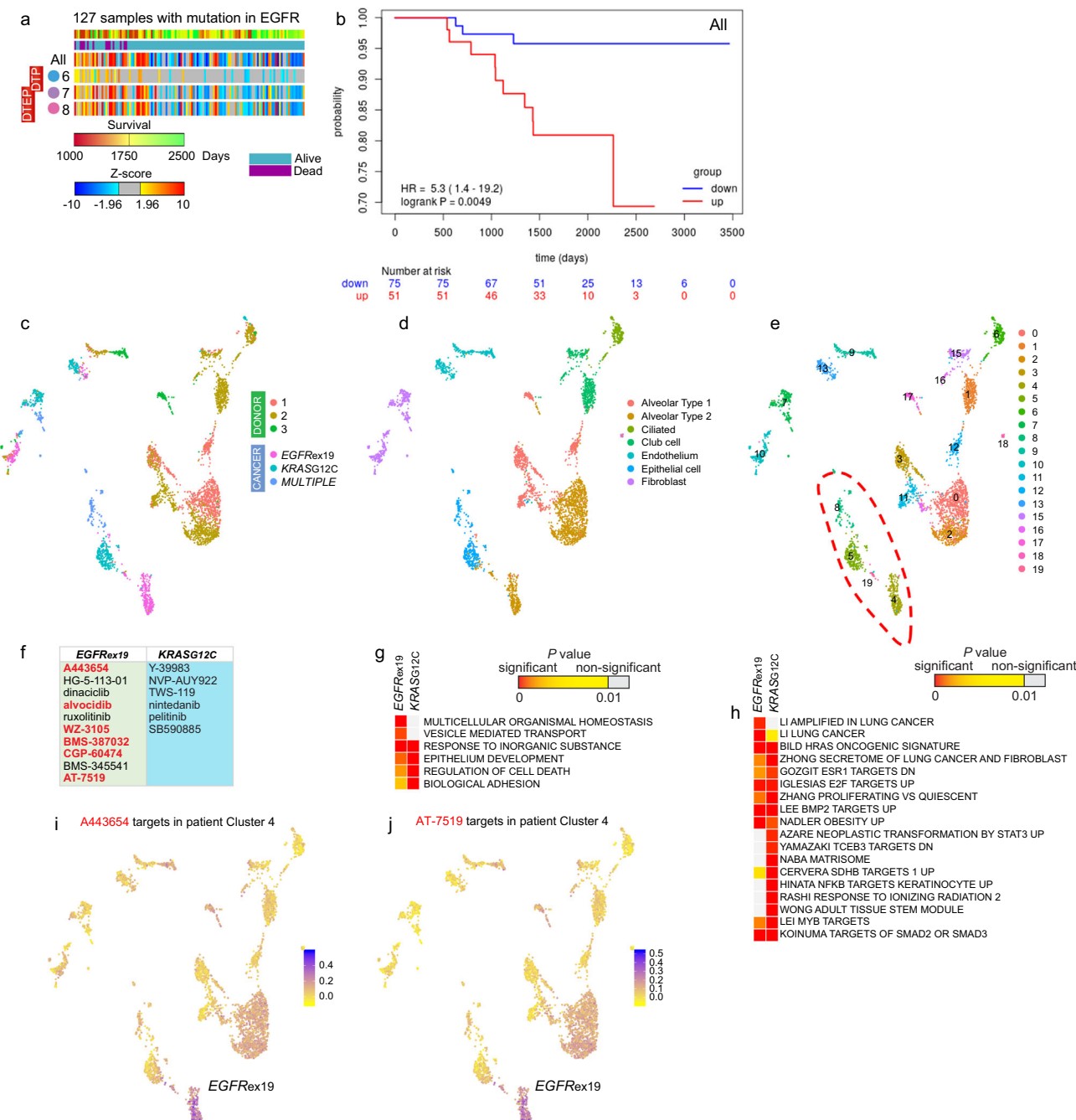

**Fig. 8 scRNA-seq identifies cancer cell populations and potential drug sensitivity in patient tumors. a** Sample-level enrichment analysis of DT markers, which were identified in the set of consecutive PC9 samples treated with erlotinib, in *EGFR*-mutant lung adenocarcinomas. Expression level was analyzed for markers of DT states in PC9 cells (Clusters 6, 7, and 8 in Fig. 1, and genes in Supplementary Data 1), for each individual DT state (6, 7, and 8) and for the three DT states altogether ("All"). **b** Kaplan–Meier estimates of survival before death/censored in 51 patients with significantly upregulated DT markers of three states (All) compared to 75 patients, where DT markers showed decreased expression or no significant change ($P > 0.05$). Univariate Cox regression was used to determine Hazard Ratio (HR) and log rank $P$ values. **c–e** UMAP representation of the epithelial subset of clusters in six tissues from different patients, colored by assigned cell types in **d**, and by cluster in **e**. Cancer cell clusters are indicated by red dotted line. **f** Top drugs identified in LINCS analysis as downregulating markers of *EGFR*ex19 and *KRAS*G12C patient tumors. Small molecules overlapping with top 10 LINCS drugs predicted for DT PC9 clusters are in red. **g** Enrichment analysis of *EGFR*ex19 cancer cluster markers ($n = 78$) and *KRAS*G12C cancer cluster markers ($n = 102$) for gene relations to GO BP terms. Terms with $P < 10^{-6}$ were included. **h** Enrichment analysis of cancer cluster markers for CGP gene sets. Gene sets with $P < 5 \times 10^{-6}$ and >6 markers were included. In **g** and **h**, data represents right tail $P$ values, two-sided binomial statistical test, adjusted for multiple testing using Benjamini-Hochberg FDR method. **i, j** Feature plots showing cells colored by expression level of genes targeted by the AKT inhibitor A443654 or CDK inhibitor AT-7519 identified in the LINCS analysis. The score was calculated by Seurat and was based on expression level of markers of *EGFR*ex19 patient cancer cells (Cluster 4 in **e** and genes from Supplementary Data 27).

Fig. 15f–i), which may be indicative of tumor dependency for predicting clinical benefit from erlotinib therapy[50].

## Discussion

The analysis described in this paper demonstrated the utility of single-cell RNA data for following purposes: (1) distinguishing drug-tolerant states; (2) discovering unique cell subpopulations, likely reflecting functional heterogeneity of drug-tolerant cells; and (3) selecting effective drugs and drug combinations to target persister cell subpopulations. We have identified cell subpopulations that are acquired early during drug treatment and do not pre-exist in the original cell population. There is a similarity in cell populations emerging after a drug holiday. Cell populations were characterized by individual markers, gene sets, biological processes, and pathways, which were not previously associated with drug tolerance. Knowledge acquired on cell populations may be used in computational prediction of drugs that work synergistically when combined. scRNA-seq revealed cell populations that have dependency on an activated bypass signaling, as it successfully predicted the targeting of a single RTK signaling effector that resulted in lethality of specific cell subpopulations. The identified markers of drug-sensitive and drug-tolerant populations could be applied clinically to identify responders to combination therapy. Through the use of scRNA-seq, EGFR-mutant tumor cells were found to be significantly different from other NSCLC tumors. Higher expression of resistance markers was significantly associated with worse patient outcome, which is consistent with the clinical data that some patients develop resistance and others never respond to targeted inhibition. These findings imply that prediction based on single-cell gene expression data will allow for the identification of treatment strategies targeting cell population heterogeneity.

In our study, the correlation between high expression of the identified tolerance markers and poor patient survival reflects the notion that aggressive tumor development is the result of a persister cell state[7]. The systematic identification of genes and cell types associated with drug resistance could advance translational science in two important ways. First, the identified (e.g., through enrichment analysis) molecular mechanisms linked to top drug tolerance genes advance our understanding of how drug tolerance emerges. Second, since we can link each cell with mechanism-related changes, we envision using this type of data to help guide screening and targeting strategies. Patients stratified by markers of tolerant cell populations can enter into clinical trials for combination therapy. In the absence of a FDA-approved drug like crizotinib, the discovery of targetable markers may be extended to drugs in preclinical and early clinical trials.

Analysis of DT markers showed that that there are individual tolerant clusters and larger cell subpopulations with distinctive molecular phenotypes. We have identified decreased cholesterol metabolism, increased EMT and tissue development, drug metabolism, and epigenetic deregulation (Table 1) as hallmarks of DT cells, which may bestow the DT cells with the significant plasticity. Consistent with previous studies, there was enrichment in gene sets activated in response to stimulus and to NF-κB and MAPK signaling. Comparison of top markers and ontology terms between cell lines with the same targeted mutation gave similar results. Our findings reflect clinically relevant biological processes, as they are consistent with identification of residual disease states. This includes cell injury and survival signals, revealed in a recent longitudinal scRNA-seq analysis of human lung tumors[51].

The identification of different drug-tolerant states by scRNA-seq suggests that drug tolerance is a continuous, multistep process that transcends previously characterized DTP and DTEP classifications. Significantly, our analysis overcomes limitations imposed by traditional methods, in which transition states were obscured in pooled populations of drug-tolerant cells expanded from persisters. This is not unprecedented, as transitioning between states has been described in response to BRAF inhibitors using single-cell proteomics[52]. We have detected several prominent biological processes and pathways in DT cell subpopulations. The identification of modules related to EMT and epithelium development was consistent with the predicted importance of EMT in driving EGFR-TKI resistance. Cells undergoing EMT are intrinsically resistant to EGFR inhibitors[53] and EMT has been associated with resistance in the clinic[54,55]. In fact, an EMT module contributing to EGFR resistance was found to be common to various NSCLC cell lines and patients[31]. Activation of TGF-β signaling was necessary and sufficient for the acquisition of mesenchymal properties{Yao:2010ir; PhD:2013jb}. Consistent with the idea that EMT represents continuum of states[56], the EMT modules INHBA$^{high}$, MET$^{high}$, and TPM $^{high}$ were distributed between several evolving cell populations. This data suggests that EMT and altered expression of developmental genes, likely resulting in loss of cellular identity, enable the cell to escape cell death by erlotinib.

scRNA-seq data demonstrated the existence of cancer cell subpopulations with differential therapeutic sensitivity, which we were able to functionally validate. Crizotinib, which was approved by the FDA in 2011 for treatment of patients with advanced NSCLC harboring ALK rearrangements, is included in more than 150 clinical trials, 84 of which are in NSCLC (clinicaltrials.gov). The cell populations that died as result of the crizotinib combination treatment were predicted to be targeted by crizotinib, while each of the cell populations that survived was not the intended target of crizotinib (Supplementary Data 21). Because of the emerging clinical utility of EGFR/MET TKI combination[57,58], understanding the underlying mechanism is important. MET amplification may be an alternative with EGFR T790M mutation, which together account for about half of acquired resistance cases[4,59]. Crizotinib-sensitive cell subpopulations featured higher expression of the *MET* gene and active TGF-β signaling (Figs. 6c–e and 7d and Supplementary Fig. 14k), suggesting that the EMT modules that we identified are functionally significant and MET is a principal mechanism of EGFR-TKI tolerance. The experiments with crizotinib suggest that resistance heterogeneity can be diminished by combining drugs that target different tolerant subpopulations. In contrast, as our single gene knockdown and knockout experiments indicate, targeting a top marker might not be effective due to survival of non-dependent subpopulations. We favor the idea that cell fate is determined by the early response to EGF which alters expression of multiple genes that work cooperatively, rather than a few specific genes[60].

Enrichment analysis of celastrol-responsive genes showed that celastrol may have sensitized PC9 cells to erlotinib through its inhibition of different clusters, those with prominent drug metabolism through cytochrome P450 and others with NF-κB activation. CYP1B1 and other cytochrome P450 enzymes may be directly related to metabolism of substances specific to resistant cells or erlotinib itself[27]. The decrease in the cholesterol metabolism that we detected in DT cells would have a profound effect on availability of the cytochrome P450 system for detoxification, cell growth, and DNA repair after erlotinib treatment. Repressing both mechanisms may be important for sensitizing cells.

Therefore, the crizotinib and celastrol data suggests that cell tolerance may be restricted by targeting predicted cell subpopulations. scRNA-seq has been applied to identify combinatorial therapeutic options in several cancers[61–64]. A common problem with combination therapy is higher toxicity than single agent therapy without a substantial improvement in efficacy. Our model is based on LINCS analysis summarizing the

transcriptional effect of the drug across multiple treatments on different cancer cell lines and at different dosages. It is conceivable that the ranking of drugs targeting DT states, which we identified through LINCS database, should be further revised based on unfavorable pharmacokinetic and pharmacodynamic interactions. Interestingly, addition of crizotinib and celastrol 48 h after the start of erlotinib treatment resulted in the greatest reduction in cell viability (Fig. 6k). This finding suggested that an increased expression of markers in the response to EGFR inhibition was required for the drug combination to achieve optimal efficiency. It will be also important to study drug effects on normal cells and scRNA-seq investigations into therapy effects on tissue cell populations[51] will contribute to this effort.

Cross-resistance between drugs[65] makes treatment even more challenging. Many studies have explored the shared survival signaling mechanisms between response to different treatments[8,66,67]. Almost any cell line, irrespective of the tissue type and kinase dependency, can be rescued from drug-induced growth inhibition by a RTK ligand[8,9]. MET, IGF-1R, and FGFR pathways are conserved across multiple cancer cell lines in response to different agents, and pharmacological targeting of these pathways may be a viable direction in combination therapy treatment. In fact, multitargeted tyrosine kinase inhibitors are an effective NSCLC treatment[5]. Some of the common features of cells that are tolerant to a variety of small molecules likely reflect a chemorefractory stem cell-like state. We have identified the high number of CDK inhibitors and PI3K inhibitors that would downregulate DT markers. A CDK2/CDK4 inhibitor would restrain cell proliferation and block conversion to DTEPs. Gefitinib-resistant PC9 cells exhibit reduced PTEN expression, leading to increased AKT phosphorylation[68], and application of a PI3K inhibitor is known to prevent TGF-β-induced EMT[69]. Based on enrichment analysis, we argue that similarities in the drug response to distinct treatments are the result of common mechanisms operating in tolerant subpopulations. In contrast, specific mechanisms may contribute to variable responses to targeted therapy in patients with identical targeted mutation and would require a personalized, more efficacious therapy. For example, a robust IFN-response in HCC827 cells and antigen presentation in tolerant xenograft tumor cells are consistent with clinical evidence of EGFR-TKIs increasing IFNγ-induced MHC class-I presentation, which may lead to enhanced recognition and lysis of tumor cells by CD8+ cytotoxic T lymphocytes[70].

The PC9 cell line model used in this study displayed many of the AT2 markers found in EGFRex19 tumor cells, including prominent cholesterol biosynthesis[71]. As restricting cholesterol and fatty acid biosynthesis is a principal mechanism of inhibiting cell growth downstream of EGFR[72], one could envision exploring the concept of pathway inhibition specifically in EGFR-mutant tumors. One important outcome of cholesterol pathway inhibition may be its effects on Hedgehog signaling since cholesterol allosterically regulates Smoothened activity and has been shown to modulate signaling range and efficiency of the Hedgehog protein[73].

The major deficiency in existing therapies is the inability to target all drug-resistant cells. Epigenetically regulated gene sets experienced some of the greatest change across four different models in this study and in the LINCS treatments. Our study and previous reports[15] also showed that loss of H3K4me3 is a prominent chromatin feature in EGFR-TKI-treated PC9. The drug holiday experiment was consistent with expectations that the non-genetic nature of drug tolerance may account for clinical cases of successful response. Previous identification of chromatin-modifying proteins HDAC9, NCOR1, MLL, and EED by loss-of-function genetic screen in erlotinib-treated cells[74] reinforces this premise. Currently available and in-development epigenetic inhibitors can be further tested based on identified markers.

The results of our study help to address pre-existing and acquired drug resistance which limits clinical usefulness of targeted and chemotoxic strategies. A recent report on over 5000 NSCLC patients sought to determine the association of broad-based genomic sequencing and survival in NSCLC[75]. The result was frustrating, as it showed no difference in outcomes of patients who had comprehensive genomic profiling of their lung cancer versus those who had only had EGFR and ALK panel testing. If one could address cell heterogeneity through a transcriptomic study, it would have improved predictive power, as our analysis showed in EGFR-mutant NSCLC for different DT states and DT cell populations. Single-cell technologies are not immediately available in the clinical oncology setting. It is conceivable that the markers identified in a limited number of patients at the cell subpopulation level will be informative in developing molecular subtypes and treatment options.

## Methods

**Cultured cell lines.** The EGFR-mutant NSCLC PC9 cell line (mutation in EGFR exon 19, ΔE746-A750, contains 8–10 copies of EGFR) was obtained from Sigma. The U937 leukemia cell line was authenticated by short tandem repeat analysis at the UIC Genomics Core cell line authentication service. M14 melanoma cell line was obtained from the Division of Cancer Treatment and Diagnosis (DCTD) Tumor Repository (NCI), and the EGFR-mutant NSCLC HCC827 (ΔE746-A750) and H1975 (T790M/L858R) were from the American Type Culture Collection (ATCC). The regular growth medium for PC9, HCC827, H1975, and M14 cells contained RPMI (CellGro, MT10040CV), 5% fetal bovine serum (FBS) (HyClone), and 1% penicillin and streptomycin (PS). U937 cells were grown in differentiation medium, RPMI, 10% FBS, PS, containing 50 nM 12-O-tetradecanoylphorbol-13-acetate (TPA) as described previously[76]. All cells were maintained in 5% $CO_2$ in a humidified incubator at 37 °C.

**Drug tolerant cell lines.** Cells obtained from vendor were maintained for not more than six passages before collection for RNA or DNA analysis. In all, $1 \times 10^6$ PC9 cells, $2.5 \times 10^6$ HCC827 cells, or $2.5 \times 10^6$ M14 cells were seeded on a p100 plate. For RNA-seq, Day 0 cells were plated 2 days before collection. Day 11 cells were plated 12 days before collection; the medium was changed every 3 days, with the 9th day being the last day. The time-course Drop-seq experiment for PC9 cells (samples D1, D2, D4, D9, and D11), which were treated for 1, 2, 4, 9, or 11 days, was initiated by adding erlotinib after the cells attached. Media for D4 was changed 48 h before cell collection. Media for D9 and D11 was changed every 3 days, but 48 h before cell collection. For generating all drug-tolerant cells, cells were continuously treated with respective drugs starting 24 h after plating, unless indicated otherwise. For a scRNA-seq experiment on the 3rd day of treatment, a respective drug was added to cells grown in 10 mL of media for 24 h, and 24 h afterwards, i.e., 48 h before cell collection for experiment, the medium was changed on fresh drug-containing medium. Untreated cells (D0) were harvested 48 h after seeding. For experiments in a p100 plate, a respective diluted drug (all prepared in DMSO) was added to cells cultured in 10 mL of media: 10 μL of 2 mM erlotinib (Thermo Fisher Scientific) were used for PC9 to obtain the final concentration of 2 μM, and 10 μL of 3 μM of erlotinib were used for HCC827 cells to obtain the final concentration of 3 nM; 10 μL of 1 mM of vemurafenib (Cayman Chemical Co) was added to each plate to the final concentration of 1 μM; and 12.5 μL of 20 mM etoposide were added to obtain the final concentration of 25 μM. In scRNA-seq experiments on the 3rd day of treatment, cell survival relative to the control was 7.2% for erlotinib-treated PC9, 7.6% for etoposide-treated PC9 cells, 5.13% for erlotinib-treated HCC827, and 4.3% for vemurafenib-treated M14 cells. For drug combination experiments, erlotinib and celastrol (Cayman Chemical Co) were used at 1 μM or 2 μM, and AEW541, crizotinib (Sigma), and GSK-1059615 (Sigma) were used at 1 μM.

**Cell separation using antibodies.** U937 cells were seeded at $2.5 \times 10^5$ cells per mL in 30 mL of differentiation medium. U937 cells were differentiated for 72 h, trypsinized and mixed with trypsinized PC9 cells at 1:1 ratio. The PC9 cells were grown in regular growth medium. Media was changed 48 h before cell collection. The mixed cells were split into three aliquots. Two aliquots were used for purification of PC9 cells by either positive selection with anti-EpCAM magnetic microbeads (human CD326, Miltenyi Biotec Cat. 130-061-101) or negative selection with anti-CD45 magnetic microbeads (Miltenyi Biotec Cat. 130-045-801). The magnetic separation was performed using MS MACS columns and MACS separator following manufacture protocols (Miltenyi Biotec). Cells without selection were kept on ice for 2 h until cells selected as EpCAM+ or CD45- were ready, and then all three were used for Drop-seq. An aliquot of each cell suspension was

fixed in 70% ethanol and analyzed by flow cytometry using anti-APC-EpCAM (Miltenyi Biotec, Cat. 130-111-000) or anti-FITC-CD45 (Miltenyi Biotec, Cat. 130-110-631) antibodies. At least 30,000 cells were acquired using Gallios Flow Cytometer (Beckman Coulter). The data was analyzed using the Kaluza Analysis software (Beckman Coulter).

**Lentiviral transductions**. The viral packaging cell line Lenti-X 293T was obtained from Clontech and maintained in Dulbecco's modified Eagle's medium (DMEM) (CellGro) and 10% FBS (HyClone). To generate lentiviruses, the Lenti-X 293T was transfected with 20 μg psPAX2 and 6 μg pMD2.G packaging plasmids (Addgene), and 15 μg viral plasmid using Lipofectamine 2000 reagent (Life Technologies) on a p100 plate. Harvesting the virus was performed at 48 and 72 h after transfection and the collected media was pooled. The virus was concentrated using Lenti-X Concentrator (Clontech) and resuspended in 1 mL of serum-free PBS to increase its concentration ten-fold. PC9 cells and HCC827 cells were seeded in PS-free growth media on six-well plates at $5 \times 10^5$ per well and $1 \times 10^6$ per well, respectively. The cells were transduced next day with lentiviruses overnight. In total, 30 min before transduction, media was changed for media containing 6 μg/μL polybrene (Sigma). Cells were treated with 25 μL of a 10x virus stock. Cells were washed twice from the viruses with PBS and recovered for 8–10 h in regular growth medium before splitting in a 96-well plate at density of $8 \times 10^3$ cells/well or $2.1 \times 10^4$ cells/well for PC9 and HCC827 cell lines, respectively. Erlotinib treatment was initiated next morning.

*Constructs*. The lentiviral constructs were used for shRNA and sgRNA to reduce levels of target genes. Three MISSION® pLKO.1-puro shRNAs were used as controls, Non-Mammalian shRNA SHC002, eGFP shRNA SHC005, and Non-Target shRNA SHC016 (Sigma). Two TACSTD2 MISSION® shRNAs (TRCN0000056419 and TRCN0000056421) and SERPINE1 MISSION® shRNA (TRCN0000331004 and TRCN0000331070) were from Sigma. sgRNA were designed (https://portals.broadinstitute.org/gpp/public/analysis-tools/sgrna-design) corresponding to the DNA strand used as a template of transcription by RNA polymerase II, to increase genome targeting[77] (Supplementary Data 29), and cloned in LRCherry2.1T backbone using the Bsmb1 site[78].

**Single-molecule RNA fluorescence in situ hybridization**. Cells were plated in 12-well plate on poly-D-lysine coated BioCoat 12 mm coverslips (Corning) at $3 \times 10^4$ cells per well. For fixation, growth medium was aspirated and cells were washed with 1 mL PBS. Cells were fixed with HistoChoice for 10 min at room temperature (RT), washed two times with PBS, and fixed again in 70% ethanol for 1 h to overnight at 4 °C. Stellaris FISH Probes were designed against *TACSTD2* RNA by utilizing RNA FISH Probe Designer (www.biosearchtech.com/stellarisdesigner). *NEAT1* Stellaris FISH Probes with Quasar 570 dye and *MALAT1* Stellaris FISH Probes with Quasar 670 dye were used as controls. Cells were hybridized with the *TACSTD2* Stellaris RNA FISH Probe set labeled with Quasar 670 dye (Biosearch Technologies, Inc.), following the manufacturer's instructions available online at www.biosearchtech.com/stellarisprotocols. Briefly, hybridization was at 37 °C for 4 h, and washing was performed for 30 min at 37 °C, 10% formamide. Slides were mounted with Vectashield and sealed.

**Immunofluorescence and immunoblotting**. For immunofluorescence (IF), cells were plated on coverslips as above and fixed in HistoChoice for 10 min at RT. Rabbit polyclonal antibodies were used for CYP1B1 (Thermo Fisher Scientific, Cat. PIPA528040, 1 mg/ml) at a 1:500 dilution. Mouse antibodies were used for TACSTD2 (DSHB, Cat. CPTC-TACSTD2-1-s, 36 μg/ml) at a 1:18 dilution and for SERPINE1 (BD, clone 41, BDB612024, 250 μg/mL) at a 1:50 dilution. Alexa Fluor 488 goat anti-rabbit and anti-mouse IgG (H + L) antibodies (Thermo Fisher Scientific) were diluted 1:200.

Protein extraction for immunoblotting was performed in EBC 450 mM lysis buffer (50 mM Tris-HCl, pH 8.0, 450 mM NaCl, and 0.5% NP-40 with proteinase inhibitors). Protein concentration was determined using BCA (Thermo Fisher Scientific). Samples were analyzed under reducing conditions on Novex WedgeWell™ 4–20% Tris-Glycine polyacrylamide gels (Invitrogen) before being transferred onto a nitrocellulose membrane (Bio-Rad). Immunoblotting was performed using the antibodies mentioned above to TACSTD2 at 0.5 μg/mL. Mouse IgG1 antibodies were used for SERPINE1 (R&D Systems Inc., Cat. MAB1786-SP) at 0.1 μg/mL, and vinculin (Sigma, Cat. V9131) at 1:15,000 as a loading control. Blots were developed using ECL. Samples for immunoblotting were generated from at least three independent cell cultures with gene knockdown or knockout.

**Microscopy and imaging**. Cells were imaged using Camera Axiocam 702 mono attached to Axio Observer 7 motorized inverted microscope (Carl Zeiss). smRNA-FISH and IF images were taken using an inverted ×40 immersion oil objective (EC Plan-Neofluar NA = 1.3 WD = 0.21 M27) with the 1.6x OptoVar turret (Carl Zeiss). ZEN 2.6 pro software supported by ZEN Module Image Analysis HW and Module Adv. Processing/Analysis HWL (Carl Zeiss) was used to create overlay, adjust brightness and contrast across images from the same experiment and add scale bars to z stack. Counting transcripts was performed by Fiji Analyze Particles program (https://imagej.net/Particle_Analysis).

The plots representing the number of transcripts were made by using the DotPlot function in R package Seurat[22,79]. A regular DotPlot was passed as a ggplot2 object to copy the structure. The information of the object was removed as a.csv file. The data was in a table format with the column values of Average Expression (avg.exp) calculate by adding the expression values of each cells and then dividing by the total number of cells. Percent Expression (pct.exp) found by counting the number of cells that had expression >0 and diving by the total number of cells. Average Scaled Expression (avg.exp.scaled) was found by dividing the average expression of the condition by the sum of the average expression of both conditions. This table was fed into the DotPlot function of Seurat, which uses color of the dots to express the Average Scaled Expression, and the size of the dots to express the Percent Expression.

For IF, image analysis was done in ZEN, and DAPI staining was used to segment individual cells. Histogram thresholds in the GFP channel were selected by centripetal expansion (starting from the nucleus) toward the cells' boundaries until intracellular signal is selected but any background is deselected. Interactive Segmentation was used to determine mean GFP fluorescence intensity per cell. Graphs were generated using GraphPad Prism version 7.

Comparison between single-cell expression values from IF data and Drop-seq data (Fig. 6i) was performed at 16 equal incremental points. Drop-seq normalized counts were at the scale from 1000 to 16,000. Similarly, 16 equal incremental points were set as a scale for smRNA-FISH.

**Cell viability assays**. For cell viability assays, we used SYTO™ 83 Orange Fluorescent Nucleic Acid Stain (Molecular Probes) unless mentioned otherwise. In all, $8 \times 10^3$ PC9 cells were seeded per well in a 96-well plate, in four replicates. In total, 10 μL of respective dilution of a drug in DMSO and medium was added per well using a multichannel pipette.

Cells were washed twice with 50 μL PBS, fixed with 50 μL of HistoChoice for 15 min, followed by staining with SYTO™ 83 (1:5000) for 15 min. The staining solution was removed, 50 μL of PBS was added, and the plate was read in the Biotek Synergy H1 plate reader with Excitation: 530/25, Emission: 590/35, Mirror: Top 570 nm, Gain: 35, Read Height: 3.5 mm, and Read Speed: Normal.

The same staining protocol was used with Hoechst 33342 (Thermo Fisher Scientific). For in vivo imaging, cells were not fixed but instead the regular growth medium was replaced on 100 μL of medium containing Hoechst (1:5,000) and incubated for 15 min. The Axio Observer 7 microscope was configured for live cell imaging and high throughput multiwell plate imaging and analysis. Cells were imaged with a ×10 (N-Achroplan 10×/0.25 Ph1 M27) or LD Plan-Neofluar ×20 long-working distance objective (NA = 0.4) Corr WD = 8.4 M27, by capturing 5 or 10 images closely to the center of each well using the Blue channel (LED-385), 65.82% light intensity, exposure time 50 ms, OptoVar 1x, Autofocus fine mode, at Axio Observer 7 microscope.

For CellTiter-Glo luminescent assay, cells were seeded as for SYTO™ 83 and analyzed according to the manufacture protocol (Promega).

For cell counts using brightfield microscopy or hemocytometer, cells were seeded in 24-well plates, and $4.3 \times 10^4$ PC9 cells was used per well. Drug-containing media was renewed at days 3, 6, and 9. Brightfield images were taken using a ×20 objective, 15% light intensity, exposure time 10 ms, OptoVar 1x, and at Axio Observer 7 microscope. For analysis by hemocytometer, cells were trypsinized, centrifuged at $300 \times g$ for 5 min, resuspended in 50 μL of media, trypan blue was added, and counted.

For analysis of dose response of PC9 DTPs and DTEPs, original PC9 cells were first treated with 2 μM erlotinib for 2, 4, 9, or 11 days, to generate DTPs (D2 and D4) and DTEPs (D9 and D11). In particular, cells were seeded in multiple p100 plates at $10^6$ cells per plate and 2 μM erlotinib was added 24 h after the seeding. Media was renewed every 3 days for D9 and D11. In all, 24 h before the timepoint of tolerance was reached, a full 96-well plate was seeded from the tolerant cells, at $8 \times 10^3$ per well in growth medium with 2 μM erlotinib. In all, 24 h after, media containing 2 μM erlotinib was removed, and new media and 10 μL of erlotinib dilution series was added to each well.

For analysis of successive drug treatment, cells were pre-treated with Erl for 48 h, and then treated with crizotinib and celastrol for another 48 h. Another treatment included all three drugs simultaneously for 96 h.

Colony formation assays were carried out in six-well plates. PC9 cells were seeded at $1.6 \times 10^4$ per well, and HCC827 cells and M14 cells were seeded at $4 \times 10^4$ cells per well. The treatment started the next day: with 1 μM erlotinib or 1 μM etoposide for PC9, with 7.5 nM erlotinib for HCC827, or with 1 μM vemurafenib for M14 cells. Additional drugs for drug combinations were added simultaneously, following concentrations in figure legends. Media was changed every 3 days. Cells were fixed with 10% of buffered formalin phosphate for 7 min, stained with 0.2% crystal violet for 30 min and pictures were taken with the Azure Bioanalytical Imaging System Model c300 (Azure Biosystems, Inc.) using the Azure Capture cSeries software version 1.97.0802 set at light grayscale UV302, exposure 1s750ms, aperture F1.4, lowest sensitivity. Colony surface area was measured using Zen Software 2.6 with Segment by global thresholding with interactive settings using the controls as initial setup.

ApoTox-Glo™ Triplex Assay (Promega) was performed following manufacture instruction and 72 h after addition of a drug. The luminescence was measured in the Biotek Synergy H1 plate reader.

**Patient tissue processing for scRNA-seq.** The study meets non-subject research because it did not recruit patients, used only de-identified tissue samples, and no identifiable data was collected. The Office for the Protection of Research Subjects (OPRS) Institutional Review Board (IRB) confirmed that the research activity does not appear to involve "human subjects" as defined in 45 CFR 46. 102(f). Each subject was assigned a unique study ID via the UICC Lung Cancer Biospecimen Biorepository, which obtained the written informed consent and for which approval from the Office for the Protection of Research Subjects (OPRS) Institutional Review Board (IRB) of the University of Illinois at Chicago was obtained. The tissue (~3–5 mm$^3$) was supplied in 5 mL of Tissue Storage Solution (Miltenyi Biotec) on ice and was loaded for the flow on Drop-seq instrument not later than 6 h after the surgery. Fat, fibrous, and necrotic areas were removed. The tumor tissue was chopped in a drop of 200 μL of the Enzyme Mix of the human Tumor Dissociation Kit (Miltenyi Biotec; 50 μL Enzyme H, 25 μL Enzyme R and 6.25 μL of Enzyme A diluted in 1119 μL DMEM) on ice and transferred using a 1000-μL Gilson pipet with cut-off tip to a 2-mL tube. Then, the sample was incubated for 15 min at 37 °C with a vigorous rotation, pipetting it up and down (3x) using a 1000-μL Gilson pipet tip (without cut-off) for 2 min every 5 min until there is no macroscopic chunks. The cell suspension was passed through a 40-μm strainer, collecting the cells by passing 20 mL of DMEM through the strainer at RT. Counting with trypan blue and immunofluorescent labeling (after cell fixation in 4% paraformaldehyde) was performed at each experimental checkpoint. After passing through the strainer, the cell suspension was centrifuged at $300 \times g$ for 7 min at RT. 0.6 mL of ACK lysing buffer (Lonza) was added to 1 mm$^3$ of pellet and the pellet was resuspended by gentle pipetting using a 1000-μL pipet tip (with a cut-off). The lysis was performed on ice for up to 2 min, after which the tube was filled with 14 mL of PBS/0.04% BSA and centrifuged at $300 \times g$ for 5 min. The ACK treatment was repeated if the pellet was still red. The cell pellet was resuspended in 300 μL PBS/0.04% BSA, cells were counted and used for scRNA-seq.

**Mice xenograft.** The mouse experiments described in this study were approved by the University of Illinois at Chicago Office of Animal Care and Institutional Biosafety Committee (OACIB; institutional Animal Welfare Assurance No. A3460.01). The study has complied with all relevant ethical regulations for animal testing and research. PC9 cells were transduced with pLenti PGK Blast V5-LUC (w528-1; Addgene) lentiviruses at ~0.6 multiplicity of infection (no selection with blasticidin). Two passages after the transduction, $5 \times 10^6$ cells were suspended in FBS-free DMEM (without phenol red) and mixed with Matrigel at 1:1 ratio, and 100 μL of this cell suspension was injected subcutaneously into the right side flank of female athymic nude (nu/nu) 6-weeks-old mice (Charles River Laboratories) as described previously[80]. The cells were free of mycoplasma. Tumor growth was measured twice weekly by bilateral calipers and once a week using the IVIS Lumina II imaging system (PerkinElmer) after injection of 100 μL 12.5 mg/mL luciferin (50 mg per kg) via IP. Tumor volume was calculated from capilar measurement using formula for the ellipsoid (0.52 × length × width$^2$). The mice have reached tumor volume between 200 mm$^3$ and 240 mm$^3$, which occurred between 10 and 15 days after the implantation. Osimertinib (AZD-9291, LC Laboratories) and crizotinib (PF02341066, LC Laboratories) were prepared in 0.5% hydroxypropyl methylcellulose and 0.5% polysorbate 80. Mice were dosed daily by oral gavage for 3 days with 0.1 ml of vehicle, the combination of osimertinib (5 mg/kg/day) and crizotinib (20 mg/kg/day), single drugs, or vehicle. Two mice were used per experiment. The mice were sacrificed and tumors were removed. Tumor dissociation was performed as for human lung tissue, and the tumor cells were loaded on the Drop-seq instrument 50 min after the mouse euthanasia. The cells were counted and used 1:1 from two experimental mice per Drop-seq sample.

## Quantification and statistical analyses
### RNA sequencing and weighted gene co-expression network analysis
*Microfluidic single-cell capture and cDNA library preparation.* Cells were trypsinized, collected in FBS-free medium by centrifugation at $300 \times g$ for 5 min, washed in PBS, 0.01% BSA, and collected again at $300 \times g$ for 5 min. The pelleted cells were resuspended in 1–2 mL PBS and filtered through a 40-μm strainer. Cell counts were obtained using Trypan blue and hemocytometer. Cell suspension was adjusted with PBS/BSA to $1.6 \times 10^5$ cells in a final volume of 1.5 mL for the flow.

*Drop-seq.* Biological replicate samples were generated to ensure reproducibility of the critical treatment points (listed in Supplementary Table 1) and analyzed as in Supplementary Fig. 12. We followed the Drop-seq protocol (Macosko et al. 2015) "Online-Drop-seq-Protocol-v.-3.1-Dec-2015.pdf" at http://mccarrolllab.com/dropseq/ while having three modifications: (1) the final concentration of Sarkosyl in the lysis buffer was 0.4%; (2) the

cDNA Post PCR was purified twice with 0.6× AMPure beads; and (3) the tagmented DNA for sequencing was purified twice: first using 0.6× AMPure beads and the second time using 1× AMPure beads. Pre and post-tagmentation libraries were checked with Agilent TapeStation 4200. We further performed sequencing on an Illumina NextSeq500, V2, High-output (400 M clusters) with custom Drop-seq read 1 primer.

*10x Genomics.* Cells were suspended in PBS/0.04% BSA targeting to generate 10,000 single cells, mixed with Master Mix and loaded onto a 10x Genomics Chromium Single Cell instrument. All the procedures were done using Single Cell 3′ Reagent Kits v3 following the 10x Genomics user guide #CG000183 Rev B. Libraries were run on the Illumina NextSeq500.

*Bulk RNA-seq cDNA library preparation.* Total cellular RNA was isolated by TRIzol (Invitrogen), which was followed by QIAGEN miRNeasy Micro purification according to the Qiagen protocol. Then, we followed Illumina TruSeq Stranded Total RNA Ribo-Zero Human library construction protocol for removal of ribosomal RNA. The RNA-seq libraries were prepared at the Whitehead Institute Genome Technology Core and single-read sequencing for 40 bases was performed on an Illumina HiSeq 2500 Analyzer.

*Processing of scRNA-seq and RNA-seq raw reads and quantification of gene expression.* Basic assessment of Illumina sequencing output reads (FastQ format) quality including GC bias was performed using FastQC program (http://www.bioinformatics.bbsrc.ac.uk/projects/fastqc/). Before mapping, poor quality reads were removed based on default quality flag in FastQ file by Illumina pipeline.

Drop-seq data was analyzed using the Drop-seq core computational pipeline described in Drop-seqAlignmentCookbookv1.2Jan2016 (http://mccarrolllab.com/dropseq/) and using the Drop-seq_tools-1.13. Mapping was performed using STAR aligner (version 2.5.2a) and sequencing reads were aligned to human reference genome/transcriptome version GRCh37.p13 using Ensembl gene model (release 74)[81]. For the 10× Genomics platform dataset, the BAM files were demultiplexed using the default parameters of the Cell Ranger v2.0 software (10× Genomics), which records the number of UMIs for each gene that are associated with each cell barcode, and human genome version GRCh37.p13 with Ensembl gene model (release 74) as the reference. The aligned libraries for Drop-seq were further processed into Digital Gene Expression (DGE) matrix using the Drop-seq program DigitalExpression tool (integrated in Drop-seq_tools-1.13). The number of cells that were extracted from aligned BAM file was based on the knee plot procedure, which extracts the number of reads per cell, then plots the cumulative distribution of reads and selects the knee of the distribution. To account for different sequencing depth across cells, the total number of UMIs per cell was considered and UMI counts were converted to transcripts-per-10,000 using Seurat[22,79] package with the following parameters: normalization. method = "LogNormalize", scale.factor = $10^4$. Because the captured erlotinib-treated cells might be particularly highly stressed, broken or dead, Drop-seq data was assessed for features of low quality cells[82]. Of particular concern was the quality of early drug-tolerant cells because the number of reads as well as of detected genes, including lower expressed genes, was higher in untreated condition. We used percent of ribosomal protein transcripts (percent.RP) as a measure of library complexity. As the cell membrane is broken down, cytoplasmic RNA is lost first, increasing the relative cellular content of mtDNA RNA. Thus, percent of mitochondrial transcripts (percent.MT) contributed to assessing cell quality. The number of highly variable genes and transcriptome variance were also indicative of cells quality. The following parameters, which included basic statistics created by FastQC (version 0.11.5) report, were used for assessing cell quality: nFeature_RNA, the number of non-exonic

RNA reads, nFeature_RNA to nCount_RNA distribution, percent.RP, percent.MT, Dispersion or Standardized Variance to Average Expression, the number of significant PCs, whether there were cells predicted to be doublets, and the identity of top cluster markers (such as no irrelevant stress response genes; Supplementary Fig. 17). The cells have passed the quality criteria above, and only nFeature_RNA range (Supplementary Table 2) was used for filtering cells. We empirically determined the cut-off for the minimum number of genes per cell (nFeature_RNA), which was not high enough to disproportionately filter out the cells but also not low enough to affect the cell distribution by inclusion of low-quality cells, while avoiding rare cell doublets by setting the nFeature_RNA_max as indicated in Supplementary Table 2.

RNA-seq reads were mapped against human reference genome/transcriptome version GRCh37.p13 (Ensembl release 74) using TopHat (version 2.1.1)[83]. After read quality assessment by FastQC program, identified short reads were uniquely aligned allowing at best two mismatches to the reference genome. Sequences that matched more than one place with equal quality were discarded. The reads that were not mapped to the genome were utilized to map against the transcriptome. Any residual mapping bias was checked using RSEQtools[84]. Consistency between two replicate data were checked by Pearson correlation co-efficient (PCC) by counting reads in 500-bp bins. After mapping, SubRead package 'featureCount' (version 2.21)[85] was used to calculate absolute read abundance (read count. rc) for each transcript associated to Ensembl gene. First strand cDNA was synthesized from random hexamers, which required read count reweighting scheme to eliminate sequencing bias[86]. Therefore, we calculated weighted counts (wc) using the R/Bioconductor[87] package Genominator (version 1.2.4). With genes >9 counts, we applied R/Bioconductor package DESeq (version 2)[88] for differential expression analysis using the negative binomial model. The data were normalized based on total mapped reads in combined analyzed samples, and using inherited functions, checked for fitting the model density of residual variance ratios. As we used two biological replicates in each condition, we calculated means of the normalized weighted counts and reported them as baseMean values (Supplementary Data 6). The MA plots were produced for overall visualization of differential expression pattern. log2FoldChange values for each gene between two erlotinib-treated and two untreated samples are reported at FDR < 0.05. The overall read mapping rate ranged from 97.2 to 97.6%, with 46,238,744 and 40,713,829 mapped for the two repeats of untreated cells, and 55,800,161 and 45,006,583 for the two repeats of treated cells. The total number of mapped transcripts was 28,669. Total 8787 genes were expressed at FDR < 0.05, and among 6659 protein coding genes, 3490 genes were upregulated, and 3169 genes were downregulated.

For comparison between Drop-seq and bulk RNA-seq or ChIP-seq data, cells were grown for all types of samples following the same protocol in duplicate. Drop-seq samples were generated and analyzed using DESeq (version 2), which finds markers using a function similar to that used in RNA-seq. UMI counts were converted using Seurat package with the following parameters: normalization.method = "LogNormalize", scale.factor = 10^7. Seurat objects were created as follows: CreateSeuratObject min.cells = 0, min.features = 1000. Only protein coding genes were included, which gave 15,965 genes for D0 and D11 samples combined. This list was also used as a background file for Gitools enrichment analyses. The threshold for considering gene differentially expressed was setup at log2FoldChange > 0.1, pct > 0.01 (which meant the gene is expressed in at least 1% of the cells in the sample). For RNA-seq sample data, the threshold for considering gene differentially expressed was setup at P < 0.05, which corresponded to log2FoldChange > 0.22. Drop-seq and RNA-seq data shared many of the leading-edge genes: nuclear protein transcriptional

regulator 1, *NUPR1*, ranked third in both data, and insulin-like growth factor binding protein 5, *IGFBP5*, ranked seventh in Drop-seq data and second in RNA-seq data.

*Identifying transcriptionally distinct cell clusters.* Extracted digital gene expression matrices were subjected to unsupervised clustering analysis using R package Seurat version 3.1.2, R version 3.5.3[22,79]. For comparison of PC9 and U937 cells, R version 3.5.0 and Seurat version 2.3.4 were used.

To confirm that we can distinguish single cells based on their transcriptomes, we profiled a mixture of two very different cell lines, the NSCLC cell line PC9 and histiocytic lymphoma cell line U937 (Supplementary Fig. 3 and 4, and Supplementary Data 9). t-SNE (t-distributed stochastic neighbor embedding) method in Seurat unbiasedly groups cells based on their gene expression profile by providing a two-dimensional embedding of single cells. Seurat analysis of the transcriptome mixture (Supplementary Table 2) separated the cells into two distinct groups (Supplementary Fig. 3c). PC analysis clearly identified and separated cells that expressed epithelial and hematopoietic markers (Supplementary Fig. 3d). Flow cytometry analysis using antibodies to an epithelial and a hematopoietic marker has confirmed the Drop-seq result related to retention of unrelated cell line after both positive and negative selection (Supplementary Fig. 3b; for gating details, see Supplementary Fig. 18). Analysis of highly differentially expressed genes allowed identifying cell-type-specific markers (Supplementary Data 9). The group with canonical epithelial markers *KRT17* and *EPCAM*[89,90] was classified as the PC9 cells (Supplementary Fig. 3e). The group with high expression level of the high-affinity IgE receptor *FCER1G* and hematopoietic cell signal transducer *HCST*[91,92] was classified as U937 cells. Cell identification by Drop-seq (Supplementary Fig. 3c) outperformed the routine method of cell separation, magnetic labeling with surface-specific antibody, in both negative and positive selection (Supplementary Fig. 3b). Drop-seq thus allowed for efficient cell separation, while positive selection with EpCAM antibody and negative selection with CD45 antibody failed to separate 1–5% and ~25% of U937 cells, respectively (Supplementary Fig. 3c).

To test the performance of Drop-seq and computational pipeline Seurat in revealing intrinsic cell subpopulations, the U937 cells were treated with differentiation agent 12-O-tetradecanoylphorbol-13-acetate (TPA) to generate a mixture of monocytes and macrophages[93]. Many of the top differentially expressed genes have been previously detected in bulk samples of TPA-induced U937 cells and other myeloid cells (Supplementary Fig. 4a, b and Supplementary Data 9 and references herein). The β2-integrin ITGAM/CD11b was previously detected using FACS and microarray as the cell surface marker of monocyte-derived macrophages[94]. Four distinct subpopulations were identified in U937 cells in an unbiased way, and were named according to canonical markers as monocytes (*IL1B* and *IL8*), macrophages (*LYZ* and *CCDC88A*), a cell cluster at intermediate state (*ALOX5AP*), and cells that failed to differentiate (Supplementary Fig. 4c, d). Thus, the accuracy of Drop-seq, i.e., encapsulation, loading quantity, and cDNA library preparation, has been validated by its sensitivity and specificity in detecting different cell types and cells transitioning between different states in a heterogeneous context. Drop-seq has allowed for identification of the intermediate state, which had not been detected by traditional methods, as well as new markers characterizing particular states.

For all samples, mitochondrial genes were removed after the mapping. For patient samples, preudogenes were removed along with the mitochondrial genes. We removed genes expressed in less than three single cells. Using genes variable in expression between the clusters as input, we performed principal component

analysis (PCA) dimensionality reduction. The Seurat function "FindClusters" was utilized to identify clusters of cells with similar transcriptomes based on their PCs. The dimensions and resolutions were dataset-dependent and are reported in Supplementary Table 2. The optimal number of clusters were estimated taking in consideration: (1) principal components (PCs) with $P$ value $< 10^{-10}$, calculated using "JackStraw" function; (2) highest average silhouette width[23] for the combined clusters within a range for resolution from 0.5 to 2, where a higher silhouette width value means the cells are in correct cluster and a lower value indicates that the cells have high enough probability to belong to another cluster; (3) avoiding appearance of different samples in the same cluster; (4) curating clusters for identical top genes (to prevent over-clustering); and (5) the list of enriched gene signatures determined using enrichment analysis, where absent enriched gene signatures would indicate under-clustering, while a heavy overlap in gene signatures between two clusters would indicate over-clustering. The later point is important for revealing non-redundant functional clusters. For the 10x Genomics dataset, in addition to these five criteria, we also used Velocyto visualization to point the changes in directionality from one cluster to another and avoid having changes in directionality within one cluster. For the patient sample dataset, the function "SCTransform" in Seurat was used to select more PCs, as stated in Supplementary Table 2. The resolution was set to highlight the biologically significant differences between cell subpopulations and is reported in Supplementary Data 2. The function "FindAllMarkers" in Seurat was used to identify clusters markers. The following computational figures: dotplots, feature plots, tSNE, UMAPs, heatmaps, and gene/gene plots were generated using Seurat. Top markers were manually selected from marker lists based on the relatively higher gene expression level in a given cell cluster, which distinguishes that population of cells from all other clusters (e.g., untreated cells (Clusters 1, 2, and 3) and treated cells (Clusters 4, 5, 6, 7, and 8) in Fig. 1g). Top marker expression levels are presented as dot plots in the figures.

Cell types were assigned to the identified clusters taking into account the abundance of canonical marker genes. Genes defined as being previously known to be cell-cycle regulated were based on Whitfield study[95]. This gene set was used for the cell cycle phase positioning of each cell. In cell line data, top principal components were dominated by genes involved in DNA synthesis, DNA replication, and other processes in the cell cycle. Thus, this gene set was subtracted before clustering for the testing if cell cycle genes are driving the heterogeneity of different DT states. In UMAP representation, cells were colored by cell cycle score attributed by Seurat package.

For transcriptome analysis of cells from different treatments localized to individual clusters or cluster groups Criz-R and Criz-S, pairwise comparison between a group of cells against another group of cells was done using function "FindMarkers", with the first group as pct.1 and the second group as pct.2. Comparisons of Erl + Criz-treated cells in Criz-R clusters versus Criz-S clusters gave similar markers as the comparison of Erl + Criz-treated cells to Erl-treated cells. Likewise, comparisons of Erl-treated cells in Criz-R clusters versus Criz-S clusters gave similar markers as the comparison of Erl + Criz-treated cells to Erl-treated cells. To test if there are markers distinguishing Erl + Criz-treated cells from Erl-treated cells within the same cluster, the cells from the two treatments were compared within the Criz-S clusters or Criz-S clusters. Within the same cluster, Erl + Criz-treated cells displayed an increase in $PLA2G16$ ($P < 10^{-18}$) and $CYP1B1$ ($P < 0.001$) levels and the lack of EMT genes (decreased keratins $KRT7$ and $KRT8$, $P < 10^{-17}$; and $KRT17$, $P < 10^{-4}$, $CST6$, $P < 10^{-12}$, $FN1$, $P < 0.0002$, $TGFBI$, $P < 0.004$) compared to Erl-resistant cells. However, there was no difference in the average values calculated

for the whole set of Criz-R markers as shown in the violin plots (Supplementary Fig. 13f). The Erl + Criz-treated cells that localized to Criz-S clusters (Supplementary Fig. 13d) exhibited the top markers of the Criz-R clusters $SQSTM1$ ($P < 10^{-10}$) and $TACSTD2$ ($P < 0.004$; Cluster 9 in Fig. 6b), while maintaining decreased expression of $KRT17$ ($P < 0.005$) and $TGFBI$ ($P < 10^{-4}$).

Single-cell pseudotime trajectory was constructed using Monocle 3 (monocle3 0.1.1, R version 3.6.2)[38]. The cds object was created using the expression matrix, cell_metadata and gene_metadata using the count slot from a Seurat object. The pre-processing step was performed with 100 dimensions using the "residual_model_formula_str" subtracting the cell cycle phase information retrieved from the meta.data of the Seurat object. Notably, clusters identified by Monocle were very similar to the clusters identified by Seurat.

We measured RNA velocity of single cells using program Velocyto (version 0.17.17)[24]. For Fig. 1e, the Velocyto function "show.velocity.on.embedding.cor" was used with default parameters: kGenes = 1 (number of genes (k) to use in gene kNN pooling), deltaT = 1 (amount of time to project the cell forward), and kCells = 10 (number of k nearest neighbors (NN) to use in slope calculation smoothing). For Fig. 4a, the function "velocyto run10x" was run on 10x Genomics BAM files from Cell Ranger software (10x Genomics) to create the "loom" files. Then, we used the SeuratWrappers function RunVelocity with default parameters.

The strength of connection was calculated between the cells in each pair of clusters using a k-nearest neighbor (knn) principle. The PAGA graph[96] was made using the preprocessed Seurat object. For this, the Scanpy[97] function scanpy.pp.neighbors was ran using the PCA embeddings calculated by Seurat. Then, scanpy.tl.paga was ran using Seurat clusters as groups and finally the plot was generated using the function sc.pl.paga_compare with the following parameters edge_width_scale = 0.5, threshold = 0.4, node_size_scale = 2.0.

**Assigning cell type identity to clusters and identification of cancer cells**. For the U937/PC9 cell mixture experiment, U937 cells were subset and re-clustered using the methods above, and with resolution = 0.25, four clusters were found. While identified in unbiased manner, the clusters were assigned to cell types using markers described in the literature: $GCN$[98,99], $TYROBP$[100], $ZFP36L1$[101], $CCDC88A$[102], $METRNL$[103], and $ITGB5$[104,105] for marcophages; $ALOX5AP$[106] and $LYZ$[107] for marcophages/monocytes; and $IL1B$[108,109], $IL8$[109,110], $CCL2$[111], $CCL3$[109], $CCL7$[109,112], and $CXCL2$[113] for monocytes.

For PC9, high number of reads was obtained for the lung epithelial marker $TTF1$/Nkx2-1, and a lower number, for the common alveolar cell markers $SP5$/SPC, $LYZ$, $LAMP3$, and $ABCA3$. Genes usually expressed only in lung development $SOX9$ and high mobility group protein $HMGA2$, were found to be profoundly decreased in drug-tolerant cells.

Patient sample cell types were assigned according the biomarkers from three previous studies[46,114]. Considering that distance between cells in UMAP is indicative of their relevance, cells from UMAP clusters located in the proximity of clusters were annotated as epithelial and AP2, were subset and re-clustered using the parameters reported in Supplementary Table 2. Cell types were assigned again as above. Genes that are preferentially expressed in a cancer cluster compared to all other cell types were found by using the Seurat function FindAllMarkers using bimod test. Comparison to the normal samples was performed by selecting the cluster corresponding to the cell type of origin, and finding the markers between the cancer cluster and the nonmalignant cluster using the Seurat function FindMarkers with default parameters.

**Gene set enrichment analysis of the CCLE data**. The whole-genome data on proteomics to RNA-seq correlation was obtained from CCLE[29]. In particular, the SCC and PCC protein to RNA level data was acquired from Table_S4_Protein_RNA_Correlation_and_Enrichments (can be found at https://gygi.med.harvard.edu/publications/ccle) and a ranked list was created from the correlation values. The GSEA method[115] was run for the enrichment analysis using a gene set comprising of markers ($P < 0.05$). Average correlation value across all markers was reported for each cluster.

**GSEA of the LINCS drug data**. The Library of Integrated Network-based Cellular Signatures (LINCS; http://www.lincsproject.org/) catalogs transcriptional responses following treatments with small molecules, thus representing a resource for investigating drugs in the drug tolerance network. For each drug, transcriptional response was calculated from a combination of all experimental treatment conditions reported for a drug (i.e., different cell lines, drug concentrations, and time of treatment) and of untreated conditions. In particular, a weight average fold change difference between treated and untreated conditions representing the drug effect in a cell line was calculated for each gene, and the genes were ranked according to their differential expression. Next, the ranked lists from different cell lines treated with the same drug were merged according to the PRL methodology[116]. The PRL procedure was designed to equally weight the contribution of each of the cell lines to the drug PRL and is based on a hierarchical majority-voting scheme, where the genes consistently overexpressed/downregulated across the ranked lists of individual cell lines will hold top positions in the PRL. Thus, for each drug in L1000 Phase 1 and Phase 2 experiments, a ranked list of genes from the top upregulated genes to the top downregulated genes (named drug Prototype Ranked List, PRL) was generated. A drug PRL thus represents a "consensus" transcriptional response to a drug.

Next, we queried PRLs from the LINSC database with markers of DT states and DT clusters to recover the drugs. We tested if the genes of interest/markers are present among the top genes (i.e., up- or downregulated by a drug) in the PRL. We used GSEA[115] as it is using a ranked list of genes such as PRL against genes of interest. GSEA calculates normalized enrichment score (NES), which signifies that the genes of interest/markers are mostly represented among the genes on the top of the PRL. Positive NES value indicates that the marker is upregulated by the drug, while a negative NES will signify that the marker will be represented among the genes downregulated by the drug. To find the top drugs downregulating DT markers, the result is presented as a ranked list of drugs with negative NES values. The drugs with the lowest negative NES are expected to downregulate DT markers in their transcriptional signature. To find the top drugs activating DT markers or a biological pathway associated with drug tolerance, the result is presented as a ranked list of drugs with positive NES values.

The lists of markers of DT states or DT cell populations were generated setting the threshold of Bonferroni corrected p-value, $P < 0.05$. For the D0-D11 time-course experiment (Supplementary Data 1), Cluster 6 did not overlap with any of the drug gene datasets, thus markers from Clusters 4, 5, 7, and 8 were used. The lists of markers of DT cell populations were obtained from 3-day-long drug treatments in four different cell models followed by Drop-seq analysis: erlotinib-treated PC9 cells (Supplementary Data 35, $P < 0.05$), erlotinib-treated HCC827 cells (Supplementary Data 41, $P < 0.05$), vemurafenib-treated M14 cells (Supplementary Data 43, $P < 0.05$), and etoposide-treated PC9 cells (Supplementary Data 45, $P < 0.05$). For erlotinib and crizotinib

combination treatment, Criz-R clusters were analyzed each individually, and Criz-S clusters were analyzed combined (Supplementary Data 20). For human tissue markers, we excluded ribosomal protein genes, which were very abundant in the external donor dataset (26 RP genes out of 64 markers of combined clusters 0 and 2 (donors 1 and 2) versus cluster 4 (*EGFR*ex19 patient)), anti-sense transcripts and HLA genes. In the epithelial subset, cluster 14 contained a few cells from one donor and was excluded from further analysis. Also, for human tissues, the presented cancer cluster markers were identified in comparison to all other non-hematopoietic cell clusters. However, the results of LINCS analysis for the markers for the cancer cluster of the *EGFR*ex19 patient identified in comparison to all clusters in our dataset were similar.

**Enrichment analysis at the level of bulk tumor samples and patient survival analysis**. In total, 127 tumor samples with EGFR mutations were selected from 226 stage I–II lung adenocarcinoma samples included in GSE31210 Affymetrix Human Genome U133 Plus 2.0 Array. Two melanoma datasets were GSE65904 Illumina Human HT-12V4.0 BeadChip array and GSE53118 stage III disease using Illumina HumanWG-6 v3.0 BeadChip array (https://www.ncbi.nlm.nih.gov/geo/query/acc.cgi), both representing a mix of BRAF- and NRAS-mutant tumors. From the third melanoma dataset, from TCGA SKCM[117], there was enough BRAF-mutant samples available for analysis, mostly from metastatic disease. Z-score was calculated from sample-level enrichment analysis (SLEA) as described earlier[45] using Gitools 1.4.10. SLEA compared the mean expression value of the genes in a marker gene set to a distribution of the mean values of random gene sets with the same number of genes. The $P$ value related to the z-score was corrected for multiple testing using Benjamini–Hochberg false discovery rate (FDR) method[118]. We considered the markers to be highly significantly upregulated in a patient tumor if Z-score > 1.96 (FDR adjusted $P < 0.05$) and showed that sample in the colors of red; the markers were significantly decreased in tumor if Z-score < −1.96 (FDR adjusted $P < 0.05$) and showed that sample in colors of blue; no significant change was depicted in gray. Survival information is shown for each patient. In addition, survival groups were generated from patients with significantly overexpressed DT markers (Z-score > 1.96, $P < 0.05$) and the rest of patients. Kaplan–Meier survival plots were generated using the log rank test ("survdiff") and Cox proportional hazards ("coxph") from R Bioconductor package[87] to calculate the significance and hazard ratios, respectively, and "survplot" for the curves.

**Functional enrichment analysis**. Functional annotation of target genes was based on curated gene sets of hallmarks, Gene Ontology (GO) terms, KEGG pathways[119], and transcription factor targets (TFT) available from MSigDB collections (v6.2 updated in July 2018)[115]. A TFT dataset comprised of genes having at least one occurrence of the transcription factor binding site (v7.4 TRANSFAC) in the regions spanning up to 4 kb around their transcription start sites. Gene expression datasets available from the literature were downloaded from the collections of chemical and genetic perturbations (CGP) MSigDB collections database v7.0 released on 20 August 2019. To determine significant overlaps between our gene lists of interest and the MSigDB collections, we used GiTools[120], which calculates the probability of overlaps that are more than expected by random chance. The background list contained all genes, for which read counts were detected in the experiment. The resulting right tail $P$ values, two-sided binomial statistical test, were adjusted for multiple testing using Benjamini–Hochberg FDR method[118]. The

gene lists of interest were generated by selecting genes with expression changes that pass the significance threshold set at $P < 0.05$. Inclusion criteria for a dataset: the size of the dataset was at least 20 members; the lowest $P$ value was set for each scRNA experiment; and for highly overlapping datasets, the dataset with the lowest $P$ value was retained. For CGP, we dismissed datasets from irrelevant cell types or conditions. Resulting $P$ values were delineated in a color coded heatmap in GiTools, where color indicated the degree of significance: highest significance (red) to least significance (orange), and non-significant values were in gray.

**Enrichment of transcription factor binding**. In addition to transcription factor datasets available from MSigDB collections, the occurrences of transcription factor (TF)-binding sites (TFBSs) in the promoter regions (from 1 kb upstream to 200 bp downstream with respect to TSS) were predicted ($P$ value cut-off of 0.01) using STORM algorithm[121] and position frequency matrices (PFM) from TRANSFAC database (professional version release 2009.4)[122]. Analysis of overrepresentation of the identified putative TF motifs on promoters of markers of drug-resistant states against all promoters as background, i.e., enrichment of transcription factor binding, was carried out using Gitools[120]. FDR corrected right tail $P$ values were used for heat map representation of enriched TFs.

**RT-qPCR**. RNA was extracted using TRIzol and Direct-zol RNA miniPrep kit (Zymo Research). Real-time PCR was performed using the SYBR Green PCR master mix and iCycler CFX96 system (Bio-Rad). Sequences of primers used in RT-qPCR are reported in Supplementary Data 29. Standard curves for primers (Integrated DNA Technologies) were created using six dilutions covering three orders of magnitude of cDNA and qPCR was performed in triplicate. The threshold cycle C[t] values from each cDNA control were averaged and used to generate the standard curve.

**ChIP-seq and data analysis**. ChIP-seq was performed essentially as described[123]. Cells were fixed in 1% formaldehyde for 10 min and sonicated 20 cycles at 60% amplitude (Branson sonicator). Two independent replicates were used and ChIP was performed with anti-H3K4me3 (EMD Millipore, Cat. 07-473) antibodies. Reference DNA was the total genomic DNA sample. After adapter ligation, DNA was PCR-amplified for 18 cycles with Illumina primers and library fragments were gel-purified. The purified DNA was captured on an Illumina flow cell for cluster generation. Library preparation was performed using ChIP-seq kit Accel-NGS 2S (Swift Biosciences, lnc). Single-end read sequencing was performed on HiSeq 2500. After FastQC quality filtering, short reads were uniquely aligned allowing at best two mismatches to the reference genome GRCh37.p13 (Ensembl release 74) using the BOWTIE program (v2.3.0)[124]. Peak detection was performed with the Spatial Clustering for Identification of ChIP-Enriched Regions (SICER) algorithm[125]. Differential enrichment of H3K4me3 in Day 11 versus untreated cells was determined using SICER program script "SICER-df.sh"[125]. Total 15,544 peaks of H3K4me3 were detected as significantly (FDR < 0.01) decreased in D11 cells compared to untreated cells, and 27,175 significant peaks were detected as significantly (FDR < 0.01) increased in D11 cells, in many cases displaying multiple peaks per annotated gene.

For comparison between ChIP-seq H3K4me3 data and RNA-seq differentially expressed genes (FDR < 0.05, log2FoldChange > 0.22) or Drop-seq markers (FDR < 0.05), differentially enriched H3K4me3 peaks (FDR < 0.01) that were mapped to gene regions

were used. Genes with increased H3K4me3 and expression level in untreated (D0) cells versus D11 cells or genes with increased H3K4me3 and expression level in D11 cells versus D0 cells were analyzed in Supplementary Fig. 1l.

**Mutant allele detection**. The amplicon for detecting $EGFR^{T790M}$ mutation was generated using $EGFR$ primers with attached linkers (underlined):

CS1_T790M_FP: <u>ACACTGACGACATGGTTCTACA</u>CACCG TGCAGCTCATCA.

CS2_T790M_RP: <u>TACGGTAGCAGAGACTTGGTCTGATG</u> GGACAGGCACTGATTT. To increase detection limit of the mutant allele, we also introduced 0.15 µM of allele-specific competitive blocker[126] that preferentially hybridizes to wide-type $EGFR$ alleles rather than to mutant alleles and inhibits amplification of the wild-type allele. The lower detection limit of the allele-specific competitive blocker method is 0.01%[126]. The PCR was performed using MyTaq HS Mix (BioLine) with 0.25 µM of each $EGFR$ primer, an initial denaturation at 95 °C for 5 min followed by 28 cycles at 95 °C for 30 s, 68 °C for 20 s for blocker binding, 60 °C for 30 s, and 72 °C for 30 s, followed by 7 min at 72 °C. The products were subjected to NGS amplicon sequencing. The results were presented as percentage or fractional abundance of mutant DNA allele to total (mutant plus wild-type) DNA alleles. The frequency of T790M allele in PC9 cells was found to be between 0.15% and 0.22%, with and without blocker. H1975 cell line was used as T790M control since it contains 64% of T790M allele, which is consistent with $EGFR$ allelic detection in a MALDI-TOF MS study[127]. The results in Supplementary Fig. 1b are reported without the blocker.

**Reporting summary**. Further information on research design is available in the Nature Research Reporting Summary linked to this article.

## Data availability

Both the raw and processed scRNA-seq data has been deposited to the database Gene Expression Omnibus (GEO) (https://www.ncbi.nlm.nih.gov/geo) in SuperSeries under the accession number GSE149383. RNA-seq data is available under GEO GSE148465. ChIP-seq data is available under GEO GSE148461. Previously published data included: bulk RNA from the lung GSE31210 and from melanoma: TCGA SKCM under accession number phs000178.v11.p8, GSE65904 and GSE53118; the donor lung tissue scRNA-seq data from GSE130148 (GSM3732848 for sample 1, GSM3732850 for sample 2 and GSM3732854 for sample 3); and LINCS [https://lincs.hms.harvard.edu/db/] database data. The whole-genome data on proteomics to RNA-seq correlation was obtained from the CCLE study[29] [https://gygi.hms.harvard.edu/publications/ccle.html]. The authors declare that all data supporting the findings of this study are available within the Article and its Supplementary Information, Supplementary files or from the corresponding author upon reasonable request. Source data are provided with this paper.

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

## Acknowledgements

We thank William R. Sellers and Konstantinos Chronis for their discussion and comments on the manuscript. We owe gratitude to the Sequencing (SQ) sub-unit and the Genome Editing (GE) sub-unit of the UIC Genome Research Core for their guidance in preparation and sequencing of genomic libraries and generating CRISPR-mediated knockout cell lines, and to the UIC Head & Neck/Thoracic Cancer Disease Team for coordinating efforts on patient sample collection. This work was supported by institutional funding (E.V.B.), and R01CA211095 (E.V.B.), and R35GM131707 (M.V.F.) from the National Institutes of Health.

## Author contributions

Conceptualization by N.L.-B., M.V.F., and E.V.B.; methodology by A.F.A., M.M.A., S.J.G., M.V.F., and E.V.B.; software by A.B.M.M.K.I., R.D.C., and C.R.-P.; validation, A.F.A., A.M.G., and E.V.B.; formal analysis by A.B.M.M.K.I., S.J.G., and C.R.-P.; investigation by A.F.A., A.B.M.M.K.I., M.M.A., C.C.G., A.E.R., A.M.G., C.R.-P., and E.V.B.; resources by K.V.-N., M.P., L.E.F., and E.V.B.; data curation by A.F.A. and A.B.M.M.K.I.; writing by E.V.B. with input from all authors; visualization by C.R.-P.; supervision by N.L.-B., M.V.F., and E.V.B.; project administration by E.V.B. All authors read, edited, and approved the manuscript.

## Competing interests

The authors declare no competing interests.
