## [Peer Review File · Nature Communications]

REVIEWER COMMENTS

Reviewer #1 (Remarks to the Author):

Understanding cellular heterogeneity within a tumor cell population and its impact in the development of drug resistance / tolerance upon treatment is of utmost importance to improve cancer patient survival. Authors nicely present the cellular states and key markers associated with erlotinib persister development in PC9 cells and partly other cell types. Further, authors translate these findings into targetable pathways that present an actionable vulnerability of residual cancer cells / drug tolerant persisters.

The authors highlight EMT as a driver of drug persistence and also suggest distinct signaling pathways that drive unique cell fates during the course of drug persistence. Further, new insight is generated using comparative analysis of gene expression profiling and the LINCS database, which is an important feature that showed relevance in patient-oriented data.

The findings are interesting. However, several issues diminish enthusiasm for the current study.

Major comments

Figure 1: State and cell cycle analysis indicate clustering of day 2 samples (green) separate from similar time points (e.g. day 1), showing a S-phase cell cycle profile and cell states associated with DTEPs (cluster 7, 8). The authors need to comment and review color code for UMAP blots indicate timepoints of samples (d0-d11), Figure 1B left-most graph?

Given the phenotypic similarities of DT states across different cell types (Figure 6), authors should indicate if non-PC9 cell derived persisters are sensitive to targeting via the same small molecules used for PC9 persisters (e.g. crizotinib and celastrol)? Overall, authors should expand LINCS NES analysis for cell lines and treatments presented in Figure 6 to allow assessment across different cell types, as this is of patient relevance.

Validation of vulnerabilities that are overlapping in PC9-derived persisters in preclinical models and patients is missing. Showing drug-mediated lethality could strongly influence the evaluation of clinical relevance.

As partly addressed by authors, PC9 cells have been described to harbor a sub-clonal population of cells showing the T790M mutation. In stark contrast to prior publications showing predominance of T790M+ clones in early resistance (PMID: 26828195, Figure 2a, 2-week Erlotinib treatment), authors point to overall similar frequency (0.2%) of T790M+ cells in untreated and d11 persisters. Authors should address the low overall rate of T790M+ clones (0.2% vs prior reported 5%) and evaluate assay performance using a positive control (e.g. H1975 cells)?

Lack of discrimination for potential genetic selection / Figure 1: Conclusions regarding predominant epigenetic rather than genetic drivers of DT markers solely due to H3K4me3 analysis seem unclear. As pointed out in Figure 1E, decrease H3K4me3 is associated with decreased scRNA expression for the majority of genes. For 50% of genes, increase in H3K4me3 similarly accounts for increased

scRNAseq expression. This points to epigenetic regulation of differential expressed genes but does not exclude genetic mechanisms contributing to the selection of drug tolerant persister cells.

Experiments targeting the technical validation of DropSeq for identifying different cell types and states in heterogeneous context (PC9:U937, and U937 differentiation) only allow limited conclusion regarding the applicability of similar methodology in context of drug persisters. This is because separate clustering of two extremely different cell types via tSNE and further sub-clustering/naming of U937 subgroups by known markers is not comparable with the discovery setting in drug tolerant persisters. And further, no subclusters were apparent for U937 cells in tSNE if not assigned by known markers.

Suppl. Figure 6: The presentation of the GiTool analysis and markers discussed in the text (RRM2, CD44, ITGA6...) is hard to follow. Markers highlighted in the text should be pointed out better, e.g. via an additional dot blot of transcript expression in the supplemental figure and their association to different gene sets should be made. The reader should be guided through the different sub-figures in a chronologic manner. Suppl. Figure 7: Similarly, for highlighted markers SERPINE1, WNT/b-cat, TGFb.

Figure3: In Figure 3B, it is hard to identify untreated and treated cells. An indication within the figure or legend may be helpful. Representation of velocity analysis is too small to read directionality. Identified cluster do not show significant differences in heatmap (3C) or GO term (3D) analysis, in particular for DTEP (II) cells. Regarding the latter, could authors address the adaptation of statistical tests to define clusters given the strong increase in cell number that is analyzed?

Authors point to the need for targeting multiple hits characteristic for DT states by using small molecules. Would genetic knockdown of targets of the indicated small molecules, e.g. MET for crizotinib or PI3K for GSK1059615, result in reduced viability of persisters and thus would specific targeting of single RTK signaling effector result in lethality in persisters?

Given the separation of cell clusters by timepoint and similarly, the differential vulnerability to crizotinib (state 7,8 = DTEP) and celastrol (state 4,5 = DTP) associated with different timepoints, could authors please comment if similar findings would result from timely detailed bulk RNAseq analysis?

Pointing to relevance in patient data (Figure 7), the separate resolution of DTP and DTEP is lacking despite the earlier indicated relevance of differential profiles of both cell states. Detailed analysis is needed regarding marker expression in patient samples and survival prediction. Since DTEP markers partially account for resistance marker and cells regain proliferative capacities, authors should highlight this and put it in perspective of survival analysis. In addition, the detailed analysis of non-cancer cells in Figure 8 could be moved to the Supplement, as the manuscript predominantly focused on the characterization of vulnerabilities in drug-tolerant cancer cells and is also highlighted in this figure.

Reviewer #2 (Remarks to the Author):

The manuscript submitted by Dr. Alexandre F. Aissa et al. on " Drug resistant changes at the level of single cells inform combination therapy " revealed the heterogeneity of a NSCLC cell line PC9, which harbor 19Del of EGFR gene. There were different RNA expression patterns along with the treatment of erlotinib, which might be associated with therapeutic sensitivity. Accordingly, they found that crizotinib might be used after resistance to erlotinib. Although the technologies are good and the analyses are sound, the biological and clinical value is limited.

First of all, osimertinib has become the preferred standard of care, particularly after the OS data from FLAURA was published and first generation TKI is much rarely used, which limits the clinical interest of the current study.

Second, lung cancer is a heterogenous disease with its biological and clinical behaviors including mechanisms underlying therapeutic resistance. Therefore, using long-cultured cell lines and small cohort of tumors wont be representative to clinical specimens.

Minor comments:

1. Drug tolerance states in PC9 cells treated with erlotinib was investigated first. 2um was only chosen in PC9 cells. Though authors explained that 2 um was the concentration achieved in patients, but whether this concentration could be also applied in cell lines was undetermined. Dose response experiment may be warranted.
2. Authors only used PC9 cell lines first and then concluded that earlier events in response to EGFR TKI were epigenetic in nature. How about other cell lines also investigated in this study?
3. Why histiocytic lymphoma cell line U937 was chosen to mix with lung cancer PC9 cell line? If authors want to show the ability of SC RNA seq to distinguish cells, another lung cancer cell line could be more convincing.
4. Several cell lines including NSCLC PC9, U937 leukemia cell line and M14 melanoma cell line were used in this study, and these cell lines were investigated in different parts of this paper. Patient survival analysis was only applied in lung cancer.
5. Line 18, the key words of 'single-cell RNA sequencing' and 'scRNA-seq' overlapped.
6. Line 100, please clarify how did you test the T790M mutation.
7. Line 112, what does the figure S1C mean?
8. Line 120, the cells might not represent erlotinib-tolerant cells in such a short time of treatment. Test the IC50 of PC9 to erlotinib at different time points.
9. Line 129, it is not rational to conclude that the scRNA-seq identified markers of DT cell populations, were associated with epigenetic rather than genetic changes.
10. Line 139, there seemed no difference of NEAT1 expression.
11. Line 150, there should be 'five clusters' (4-8) for drug tolerant states of DTPs and DETPs.
12. Line 152, why do you claim 'it is unlikely that the Cluster 4 cells represent a mixture of sensitive and resistant cells'?
13. Line 170-197, this section has no relationship with the main text, delete or put it into supplementary information.
14. Line 284, which figure showed the resistance clusters in PC9 and HCC827 cell lines?
15. Line 415, how to define the low/high marker expression?

16. Figure 7C, please claim the meanings of 'ASK428, ASK440, ... , EGFRex19' in the figure legend?
17. Line 428, should be 13 different clusters.
18. Page 39, in the 'RT-qPCR' section, please list all the primers of genes you detected.

Reviewer #3 (Remarks to the Author):

In this paper, Aissa et al exploit single-cell RNA-sequencing from cell lines as well as from patient-derived samples to explore heterogeneity in lung cancer and show how the results can be used (i) to investigate the mechanisms of drug tolerance and (ii) to design drug combination therapy.

The approach they use is interesting and, in addition to the computational analyses, they validate several of the conclusions that they make.

While there is an impressive amount of datasets being collected, the rationale for including them and the conclusions drawn from the analyses is not always clear. Clarity could be much improved and more insights might even be possible if some of the datasets were better integrated with one another.

Below my specific comments:

****Major Comments****

1. Line 116-117: "Regressing out cell cycle genes returned similar clusters (data not shown), indicating cell cycle signature did not drive the distribution." This is an interesting piece of analysis that should be reported, as I would imagine that cell cycle might be driving quite a lot of the variability observed at least in the untreated cells. Also in the treated cells, it's not clear why cells from D4, D9 and D11 are sitting between D1 and D2, in the UMAP (Fig 1B); unless this is driven by cell cycle.
2. Line 120-121: "From the computational point of view, genes that were not found in the literature appear to be as functionally important as known drug tolerance genes." What does "functionally important" mean here? and how do the authors assess it computationally?
3. Line 125-126: isn't the result of the comparison between bulk RNA-seq and Chip-Seq trivial and simply due to the association of H3K4me3 with active genes? Wouldn't DNaseq be necessary to draw the conclusion on Line 128-129 about the epigenetic vs genetic changes (to prove the absence of any DNA mutations)?
4. Line 155: RNA velocity here could help assessing whether cells in cluster 4 really represent a "transitional state to DTPs".
5. Figure 3A: what's the unit on the y axis? were genes normalized before being compared? if not, this result might be driven by a handful of highly expressed genes
6. Figure 3B: a UMAP with cells colored by day of treatment should be added
7. The 10x dataset seems very rich, with multiple cell subpopulations identified, but there's no attempt to interpret those in the paper. Do they represent meaningful cell states or is it just overfitting? How do they compare to the clusters identified in the Dropseq dataset? The authors could try to use one of the many existing algorithms to estimate the optimal number of clusters (eg, gap statistic, silhouette width, etc; this also applies to the other scRNAseq dataset discussed).

8. How were the two subpopulations of resistant cells (I and II) defined? Was RNAvelocity used? It seems particularly difficult to use RNA velocity in this case, as the signal appears fairly weak (Figure 3B, bottom) - or maybe it's a problem of visualization? Projecting the cells on top of eg a PAGA graph (<https://genomebiology.biomedcentral.com/track/pdf/10.1186/s13059-019-1663-x>) would help.
9. Line 281-288: is this Dropseq data on D3 in different cell lines not shown/discussed? It seems like the only result shown is some enrichment analysis of Figure S8, but if the authors want to use it, they should include further details, as was done for the other scRNAseq datasets (QC, clustering, UMAP/tSNE visualization, etc).
10. Line 290 - 295: which dataset is this referring to? the dropseq dataset mentioned in the paragraph above or some previous dataset?
11. Line 339-340: are the resistant vs sensitive populations in any way related to the subpopulations I vs II (or any other subpopulation) in the 10x dataset? This is also important for the claim made in lines 366-368.
12. Line 413: "when compared to a random dataset of the same size": what does it mean random? how was it randomized?
13. Important details are missing/unclear in the Methods section:
- regarding the QC of the scRNAseq data: which criteria were used to select good/bad quality cells (ie, threshold on number of reads, genes, fraction of reads from mtDNA, etc)?
 - Line 860-861: which analysis is this referred to? what cell populations were being compared?
 - Line 853: "The data were normalized based on total mapped reads in combined analyzed samples". This sentence is unclear. Do the authors mean that the data were normalized by dividing each cell by the total number of reads (ie, RPM normalization)? or else?
 - Line 875-876: "Drop-seq data was assessed for features of low quality cells of both biological and technical nature". What does this mean exactly? What kind of assessment was performed and what were the results?
 - 893: "The resolution was set to highlight the biologically significant differences between cell subpopulations". This is also unclear: which biologically significant differences are the authors referring to, and how were they used to choose the resolution?

****Minor Comments****

- * Line 110-111: technically, UMAP is a visualization tool and it shouldn't be used to find cell populations, which are defined by clustering.
- * Line 167: typo, Figure 2SI -> S2I

We thank the reviewers for their constructive criticism, which allowed to improve our manuscript. We appreciate that the reviewers carefully read the manuscript – this was a lot to take in. We have added several experiments and revised both the text and figures (updated numeration is used in this response) that address all reviewers' concerns. We respond to each Reviewer below:

AUTHOR RESPONSE

Reviewer #1:

Understanding cellular heterogeneity within a tumor cell population and its impact in the development of drug resistance / tolerance upon treatment is of utmost importance to improve cancer patient survival. Authors nicely present the cellular states and key markers associated with erlotinib persister development in PC9 cells and partly other cell types. Further, authors translate these findings into targetable pathways that present an actionable vulnerability of residual cancer cells / drug tolerant persisters.

The authors highlight EMT as a driver of drug persistence and also suggest distinct signaling pathways that drive unique cell fates during the course of drug persistence. Further, new insight is generated using comparative analysis of gene expression profiling and the LINCS database, which is an important feature that showed relevance in patient-oriented data.

The findings are interesting. However, several issues diminish enthusiasm for the current study.

Major comments

Figure 1: State and cell cycle analysis indicate clustering of day 2 samples (green) separate from similar time points (e.g. day 1), showing a S-phase cell cycle profile and cell states associated with DTEPs (cluster 7, 8). The authors need to comment and review color code for UMAP blots indicate timepoints of samples (d0-d11), Figure 1B left-most graph?

We apologize for an error in Figure 1B with color labeling of samples according to the day of treatment. In the original output file, the legend listed Day 11 after Day 1: Day 0, Day1, Day 11, Day 2, Day 4, Day 9. We attached the original output from Seurat analysis that confirms the color scheme:

We have now shifted the names in the legend, so it is correct.

Given the phenotypic similarities of DT states across different cell types (Figure 6), authors

should indicate if non-PC9 cell derived persisters are sensitive to targeting via the same small molecules used for PC9 persisters (e.g. crizotinib and celastrol)?

The non-PC9 cell derived persisters that we studied were HCC827 persisters and M14 persisters, and we analyzed the effects of crizotinib, celastrol and GSK1059615. We have reported treatment of HCC827 cells that carries the identical EGFR mutation as PC9 and thus highly sensitive to erlotinib. Both PC9 and HCC827 were analyzed in 3-day long proliferation assays, in apoptosis assay, and in colony formation assays (Figures 5F and G, Figures S11B and E). Colony formation assays are now performed for crizotinib, celastrol and GSK1059615 in two additional models (Figures S11F and G).

We changed text to improve the description of existing data and to introduce new data:

Main text: “LINCS analysis identified celastrol as a common top drug and crizotinib as a less significant drug, downregulating DT markers in two other drug tolerance models (Tables S14c-f, and data for crizotinib not shown). Celastrol and crizotinib were confirmed in survival assays, in contrast to GSK1059615, which was not potent in decreasing cell survival in PC9 Eto or M14 Vem (Figures S11F and G), which is consistent with the fact that these models are not known to be not dependent on activated AKT pathway for survival.”

Figure S11 legend: “(F) and (G) Colony formation assays of PC9 and M14 cells, respectively, treated for indicated number of days with Eto (etoposide) or Vem (vemurafenib) and other drugs. Celastrol was used at 1 μ M concentration. Representative crystal violet stainings are shown. Plate colony surface area is shown as mean \pm s.d. for $n = 3$ replicate wells, two-tailed p values were determined by unpaired t test relative to the DMSO control or Erl via GraphPad Prism 7. In (C) through (G), (ns) $P > 0.05$; (*) $P < 0.05$; (**) $P < 0.01$; (***) $P < 0.001$, (****) $P < 0.001$. In (B) through (G), the drug concentrations are as in Figure 5F unless noted otherwise.”

Methods: “PC9 cells were seeded at 1.6×10^4 per well, and HCC827 cells and M14 cells were seeded at 4×10^4 cells per well. The treatment started the next day: with 1 μ M erlotinib or 1 μ M etoposide for PC9, with 7.5 nM erlotinib for HCC827, or with 1 μ M vemurafenib for M14 cells. Additional drugs for drug combinations were added simultaneously, following concentrations in figure legends.”

Overall, authors should expand LINCS NES analysis for cell lines and treatments presented in Figure 6 to allow assessment across different cell types, as this is of patient relevance.

We performed the analysis for four cell models, reported top 30 drugs for each DT cluster in Tables 14b-e, and discussed the common drugs in the text.

Main text: “Consistent with LINCS identification of all three drugs targeting specific clusters in both PC9 and HCC827 cells (Tables S14c and S14d, and data for crizotinib not shown), their combination with erlotinib was effective in cell survival assays (Figures 5F and G, and Figures S11B and E). Thus, we have validated crizotinib, celastrol and GSK1059615 as combination agents with EGFR TKI”. Also, see response to the previous point. Thus, we concluded analysis for PC9/erlotinib, PC9/etoposide, HCC827/erlotinib, M14/vemurafenib that are the models presented in former Figure 6.

New results are presented in Supplemental Tables:

Table S14b. Top 30 drugs identified in LINCS analysis as downregulating genes that are the top upregulated in PC9 cells treated with erlotinib for 11 days compared to untreated as identified by bulk RNA-seq.

Table S14c. Top 30 drugs identified in LINCS analysis as downregulating markers of PC9 cell populations tolerant to erlotinib.

Table S14d. Top 30 drugs identified in LINCS analysis as downregulating markers of HCC827 cell populations tolerant to erlotinib.

Table S14e. Top 30 drugs identified in LINCS analysis as downregulating markers of PC9 cell populations tolerant to etoposide.

Table S14f. Top 30 drugs identified in LINCS analysis as downregulating markers of M14 cell populations tolerant to vemurafenib.

Validation of vulnerabilities that are overlapping in PC9-derived persisters in preclinical models and patients is missing. Showing drug-mediated lethality could strongly influence the evaluation of clinical relevance.

We now report functional validation of identification of crizotinib as a drug targeting specific cell populations and described their molecular functions, in xenograft model. We also extend our study of tolerance markers in patients and report patient survival using markers of DT states in Figure S15A, and patient survival using markers of DT cell populations from our new xenograft study in Figures S16A and B.

As partly addressed by authors, PC9 cells have been described to harbor a sub-clonal population of cells showing the T790M mutation. In stark contrast to prior publications showing predominance of T790M+ clones in early resistance (PMID: 26828195, Figure 2a, 2-week Erlotinib treatment), authors point to overall similar frequency (0.2%) of T790M+ cells in untreated and d11 persisters. Authors should address the low overall rate of T790M+ clones (0.2% vs prior reported 5%) and evaluate assay performance using a positive control (e.g. H1975 cells)?

In new Figure S1B, we report the percentage of T790M mutation in PC9 cells, untreated and D11 persisters, in comparison to H1975 cell line. This analysis confirmed that Day 11 DTEPs do not exhibit an increase in the frequency of the T790M allele. This result is not new as it was first described by Sharma et al., 2010, Cell, where they most famously expressed the concept of DTPs and DTEPs. The result also matches the data by Hata et al, 2016 Nat Med, cited by the reviewer, which represents the most detailed analysis of emerging T790M during developing resistance in PC9 cells published so far. There are no other published data that would be contradictory to our result.

We attached a table that shows that we sequenced by NGS to significant depth to consistently detect a C to T transition. We also present a dilution series of the PC9 with H1975 cells (D0:T790 control) to show that our method is sensitive enough to detect 0.2% mutant allele. The percentage of each allele is shown in the lower rows of the table. Of note, H1975 (second passage after receiving from ATCC) shows 2:1 ratio of T790M to WT allele. We compared our values for H1975 with results of a MALTI-TOF MS, which has detection limits of 0.4% to 2.2% in detecting EGFR mutations (Su et al., Journal of Clinical Oncology 2012). H1975 showed both T790M and WT in MALDI-TOF MS data, which is consistent with our detection.

Experiment 2. Frequency of T790M allele in PC9 cells as determined by PCR-assisted sequencing

	Position	A	C	G	T
PC9 Day0 (D0)	581	32	45888	8	73
PC9 Day11 (D11)	581	33	53293	3	91
T790M control	581	82	17819	21	32056
D0:T790M control (50:50)	581	103	44635	20	21303
D0:T790M control (90:10)	581	43	39658	7	3226
D0:T790M control (95:5)	581	31	40954	13	1979
D0:T790M control (98:2)	581	49	56301	23	1345
D0:T790M control (99.8:0.2)	581	44	55913	18	390
PC9 Day0 (D0)	581	0.07%	99.75%	0.02%	0.16%
PC9 Day11 (D11)	581	0.06%	99.76%	0.01%	0.17%
T790M control	581	0.16%	35.65%	0.04%	64.14%
D0:T790M control (50:50)	581	0.16%	67.57%	0.03%	32.25%
D0:T790M control (90:10)	581	0.10%	92.37%	0.02%	7.51%
D0:T790M control (95:5)	581	0.07%	95.29%	0.03%	4.60%
D0:T790M control (98:2)	581	0.08%	97.54%	0.04%	2.33%
D0:T790M control (99.8:0.2)	581	0.08%	99.20%	0.03%	0.69%
			WT		T790M

Also, in the Table below please see for reference Experiment 1 from our previous submission. Each sample was run two times, with blocker (see methods) and without blocker, and sequenced deep enough to detect the mutant allele. Cells for Experiments 1 and 2 were grown independently, and the results show reproducibility in detecting both the WT and T790M allele.

Experiment 1. Frequency of T790M allele in PC9 cells as determined by PCR-assisted sequencing

	Position	A	C	G	T
PC9 Day0 with Blocker	581	24	71068	17	155
PC9 Day11 with Blocker	581	19	59619	18	112
PC9 Day0	581	15	60077	8	102
PC9 Day11	581	13	61416	8	116
PC9 Day0 with Blocker	581	0.03%	99.72%	0.02%	0.22%
PC9 Day11 with Blocker	581	0.03%	99.75%	0.03%	0.19%
PC9 Day0	581	0.02%	99.79%	0.01%	0.17%
PC9 Day11	581	0.02%	99.77%	0.01%	0.19%
			WT		T790M

We have now better described how we determined fractional abundance of T790M cells in Methods section.

Sharma et al. described PC9-derived DTEPs, which “have not acquired the *EGFR T790M* mutation or *MET* gene amplification often associated with acquired EGFR TKI resistance in NSCLC patients (Figure S1C and data not shown), suggesting a distinct state of drug insensitivity.” For analysis of *EGFR T790M* mutation, they PCR-amplified exon 20 in *EGFR* from several tolerant clones and used HPLC analysis of the DNA restriction digest. We have used a more sensitive PCR-based approach similar to that used by Hata and co-authors. Their paper provided multiple evidence that the emergence of T790M resistant clones occurs late during the treatment. Our results matched Hata’s data: 1) in Figure S6, Hata et al showed the fraction of T790M cells in the total population and they concluded that “Intermediate resistant PC9 cells expanded from drug tolerant cells (after 12-16 weeks gefitinib treatment) are

negative for T790M by allele specific qPCR.” 2) they then performed time course experiment of the whole cell population where they say “**No T790M was detected prior to 5 months (threshold for detection 1.5%), whereas 6% of population was T790M positive at 5.5 months.** 3) in Figure S8: “**No early T790M resistant clones were observed after** treating ten independent PC9 single cell sub-clones (5,000 cells/well) with gefitinib for **two weeks.**” This reviewer probably used the value of 5% T790M from the experiment where Hata et al. cultured 1,200 small pools (5,000 cells each) of parental PC9 cells in the presence of gefitinib, from which 5-10% gave a large colony. They then sequenced 50 such quickly proliferating colonies and detected T790M mutation at similar fractional abundance as in the fully resistant PC9 clone. However, those 50 colonies represented a small percent (and not 5-10%) of the whole tolerant cell population as they showed in the experiments mentioned above.

In conclusion, our result is consistent with the published data that the T790M does not appear after 11 days of treatment yet, and there was no T790M clone that would be comparable to the size of a subpopulation identified by scRNA-seq. To reflect the fact that only late resistant cells appear as T790M positive, we have changed the sentence “The T790M mutation has been reported for late drug tolerant expanded persisters (DTEPs) as either pre-existing or developing during drug treatment” to “One of the mechanisms explaining the emergence of eventually resistant clones was attributed to the T790M “gatekeeper” mutation in EGFR, which has been reported either pre-exists or develops after several months of continuous treatment. We confirmed, consistent with previous reports (Sharma:2010, Hata:2016), that the T790M mutation was not enriched in the initial emerging PC9 DTEPs”.

Our changes in the manuscript:

Main text: “We confirmed, consistent with previous reports ^{7,15}, that the T790M mutation was not enriched in the initial emerging PC9 DTEPs as its frequency remained at around 0.2% at Day 11 of treatment (Figure S1B).”

Figure S1 legend: “(B) Allelic frequencies in untreated PC9 cells (D0) and cells treated with 2 μ M erlotinib for 11 days (D11) are compared to that of the EGFR-T790M mutant H1975 cells (T790M control). The different dilutions of PC9 with the H1975 control show the level of detection goes to <0.2%, which is the frequency of detecting T790M allele in PC9 cells.”

Methods: “H1975 (T790M/L858R) were from the American Type Culture Collection (ATCC). ...The frequency of T790M allele in PC9 cells was found to be between 0.15% and 0.22%, with and without blocker. H1975 cell line was used as T790M control since it contains 64% of T790M allele, which is consistent with EGFR allelic detection in a MALDI-TOF MS study ¹¹³. The results in Figure S1B are reported without the blocker.”

Lack of discrimination for potential genetic selection / Figure 1: Conclusions regarding predominant epigenetic rather than genetic drivers of DT markers solely due to H3K4me3 analysis seem unclear. As pointed out in Figure 1E, decrease H3K4me3 is associated with decreased scRNA expression for the majority of genes. For 50% of genes, increase in H3K4me3 similarly accounts for increased scRNAseq expression. This points to epigenetic regulation of differential expressed genes but does not exclude genetic mechanisms contributing to the selection of drug tolerant persister cells.

We fully agree that genetic drivers can be contributing to driving cell tolerance. We checked only a single mutation, in EGFR. As we found robust gene expression changes and H3K4me3

enrichment between different DT subpopulations, we focused our work on this, previously undescribed, phenomenon. We have corrected this.

Experiments targeting the technical validation of DropSeq for identifying different cell types and states in heterogeneous context (PC9:U937, and U937 differentiation) only allow limited conclusion regarding the applicability of similar methodology in context of drug persisters. This is because separate clustering of two extremely different cell types via tSNE and further sub-clustering/naming of U937 subgroups by known markers is not comparable with the discovery setting in drug tolerant persisters.

The goal of our cell mixture experiment was to distinguish single cells based on their transcriptomes (Lane 170). Since we assembled the Drop-seq instrument ourselves, we have run a number of cell mixes, containing species mixed libraries as in the original publication by Macosko et al., 2015, Cell. RNA degradation, cell doublets, contamination from ambient RNA, computational problem would affect our ability to cluster cells into subpopulations and identify specific cell population markers. For example, ambient RNA is a concern because DT markers would be detected across all cell populations of the sample. We thought that differentiating U937 cells is a reasonable choice in this respect because the mixture of monocytes and macrophages would offer an opportunity to test the sensitivity of our computational parameters in distinguishing cell subpopulations that originate from the same progenitors. We first identified the U937 subpopulations in discovery setting, and then compared the result to the populations assigned using defined markers. Because these experiments validate the methodology, the text describing those has been moved to the Methods.

And further, no subclusters were apparent for U937 cells in tSNE if not assigned by known markers.

We thank the reviewer for providing us with the opportunity to edit the text so that our intention is clear. The clustering for U937 was performed in an unbiased way, without previous knowledge of cell-type specific markers. However, it became apparent that many known markers of monocytes and macrophages are among the top identified cluster markers (Table S9) as evident from the visualization of a plot of the expression of canonical markers (Figure S4C). Now we have changed the Figure S4C and S4D legends, to reflect the fact that while identified in unbiased manner, “U937 cell clusters correspond to macrophages and monocytes.”

Suppl. Figure 6: The presentation of the GiTool analysis and markers discussed in the text (RRM2, CD44, ITGA6...) is hard to follow. Markers highlighted in the text should be pointed out better, e.g. via an additional dot blot of transcript expression in the supplemental figure and their association to different gene sets should be made. The reader should be guided through the different sub-figures in a chronologic manner. Suppl. Figure 7: Similarly, for highlighted markers SERPINE1, WNT/b-cat, TGFb.

New dot blots have been created as Figures S7G, S9E and S14H.

Figure3: In Figure 3B, it is hard to identify untreated and treated cells. An indication within the figure or legend may be helpful. Representation of velocity analysis is too small to read directionality. Identified cluster do not show significant differences in heatmap (3C) or GO term (3D) analysis, in particular for DTEP (II) cells. Regarding the latter, could authors address the adaptation of statistical tests to define clusters given the strong increase in cell number that is

analyzed?

We have now included a UMAP with color labeling of samples. The arrows in the Velocyto projection were extended. A PAGA graph has been added to reflect connectivity between different clusters, which together with two patterns of MSigDB signatures in Figure 4C, clearly separates subpopulations I and II. Silhouette analysis is now included to support the identified clusters in Figure S8. Please see response to Reviewer #3 below.

Authors point to the need for targeting multiple hits characteristic for DT states by using small molecules. Would genetic knockdown of targets of the indicated small molecules, e.g. MET for crizotinib or PI3K for GSK1059615, result in reduced viability of persisters and thus would specific targeting of single RTK signaling effector result in lethality in persisters?

We thank the reviewer for highlighting this point. We have added relevant sentences in Discussion section. The cell populations that died as result of the crizotinib combination treatment were predicted to be targeted by crizotinib, while each of the cell populations that survived was not the intended target of crizotinib (the LINCS list of drugs per each cluster in Table S17, with crizotinib highlighted). Previous studies using EGFR inhibitors in combination with knockdown of genes in bypass/downstream signaling pathways showed reduced viability in EGFR-mutant cell lines. However, the knockdown approach is not particularly effective for activated kinases and targeting of an RTK signaling effector has been achieved using pathway inhibitors. If interested in knockdown studies, please see below:

- MET inhibition using specific siRNA-mediated knockdown, in combination with erlotinib was shown to be cooperative in downstream signaling inhibition {Tang:2008je}.
- Two different MET siRNAs decreased cell survival when using BIBW2992 (irreversible EGFR inhibitor) {QU:2014cd}.
- PIK3CA siRNAs have been shown to downregulate AKT, the downstream effector of PI3K that contributes to EGFR TKI drug tolerance. I am not particularly aware of a relatively high-profile study of PI3K or AKT knockdown in targeting TKI persisters. Besides technical difficulties with shutting down the PI3K/AKT/mTOR pathway using a knockdown approach, the complexity of the pathway results in numerous signaling feedback loops and activation of compensatory pathways to avoid the PI3K inhibition. PI3K inhibition induces a spike in glucose and that induces high level of insulin and it is pro-tumorigenic in animal model. Overall, one of the common problems with PI3K/mTOR inhibitors are cytotoxic effects and lack of significant activity.

Given the separation of cell clusters by timepoint and similarly, the differential vulnerability to crizotinib (state 7,8 = DTEP) and celastrol (state 4,5 = DTP) associated with different timepoints, could authors please comment if similar findings would result from timely detailed bulk RNAseq analysis?

We added this data in Table S14b. We report that “Celastrol and GSK1059615 were identified using data from bulk RNA-seq, but the list of identified targeting drugs was much shorter than using cluster analysis from scRNA-seq and was missing crizotinib”.

Pointing to relevance in patient data (Figure 7), the separate resolution of DTP and DTEP is

lacking despite the earlier indicated relevance of differential profiles of both cell states. Detailed analysis is needed regarding marker expression in patient samples and survival prediction. Since DTEP markers partially account for resistance marker and cells regain proliferative capacities, authors should highlight this and put it in perspective of survival analysis.

We have now shown survival analysis with markers of each individual cluster, besides “All” markers, and added this data in Figure 8A. While the DTEP (Cluster 8) markers show very highly significant difference in survival, the DTP (Cluster 6) markers, which lack cell proliferation genes, show significant difference (HR = 4.9 (1.5-16.1), P value = 0.0034) (Figure S15A). This discussion has been incorporated in the text.

In addition, the detailed analysis of non-cancer cells in Figure 8 could be moved to the Supplement, as the manuscript predominantly focused on the characterization of vulnerabilities in drug-tolerant cancer cells and is also highlighted in this figure.

Figures 7C and D have now become Figures S14A and B.

Reviewer #2 (Remarks to the Author):

The manuscript submitted by Dr. Alexandre F. Aissa et al. on " Drug resistant changes at the level of single cells inform combination therapy " revealed the heterogeneity of a NSCLC cell line PC9, which harbor 19Del of EGFR gene. There were different RNA expression patterns along with the treatment of erlotinib, which might be associated with therapeutic sensitivity. Accordingly, they found that crizotinib might be used after resistance to erlotinib. Although the technologies are good and the analyses are sound, the biological and clinical value is limited.

First of all, osimertinib has become the preferred standard of care, particularly after the OS data from FLAURA was published and first generation TKI is much rarely used, which limits the clinical interest of the current study.

We have added osimertinib model to increase the clinical interest of the study (see the response to the main criticism raised by the Editor above).

Second, lung cancer is a heterogenous disease with its biological and clinical behaviors including mechanisms underlying therapeutic resistance. Therefore, using long-cultured cell lines and small cohort of tumors wont be representative to clinical specimens.

We share the concern. With due consideration to differences between cell lines and clinical samples, the knowledge about the main biological pathways involved in resistance came from studies in cell lines (Wilson et al., Nature 2012, Bhang et al., Nature Medicine 2015, Hata et al., Nature Medicine 2016, Ramirez et al., Nature Communications 2016, Lee et al., Cancer Cell 2018). The fact that we identified crizotinib, which is now used as a combination treatment with osimertinib, as the top drug suggests clinical relevance of markers of cell line populations. While this article was in review, we were excited to learnt about scRNA-seq performed in NSCLCs from patients treated with targeted therapy (Maynard et al. Cell 2020). Strikingly, there was a large overlap in genes and processes that they associated with resistance with our study. In particular, they also focused on *SERPINE1*, the top gene in our study. Unfortunately, we are not aware of larger cohorts of EGFR-mutant NSCLCs available with gene expression and survival data.

Minor comments:

1. Drug tolerance states in PC9 cells treated with erlotinib was investigated first. 2um was only chosen in PC9 cells. Though authors explained that 2 um was the concentration achieved in patients, but whether this concentration could be also applied in cell lines was undetermined. Dose response experiment may be warranted.

We now report an IC50 experiment in Figure 1A.

2. Authors only used PC9 cell lines first and then concluded that earlier events in response to EGFR TKI were epigenetic in nature. How about other cell lines also investigated in this study?

We compare all four models in Figure 3C, which shows P values for genes in gene signatures **MISSIAGLIA REGULATED BY METHYLATION UP, HELLER HDAC TARGETS SILENCED BY METHYLATION UP, NUYTEN EZH2 TARGETS UP** across DT populations in four models. Epigenetic signatures are also now presented in the PC9 xenograft study (Figures S14E and L). In Discussion, we now say "Epigenetically-regulated gene sets experienced some of the greatest change in four different models in this study". We now changed in the main text : "a majority of highly enriched gene sets were common in all four models: EMT, tissue development, vesicle-mediated transport and epigenetic regulation (Figures 3B and C)." Reference to Figures S14E and L has been added to Table 1 cell "EPIGENETIC REGULATION".

3. Why histiocytic lymphoma cell line U937 was chosen to mix with lung cancer PC9 cell line? If authors want to show the ability of SC RNA seq to distinguish cells, another lung cancer cell line could be more convincing.

We chose PC9 since most of the experiments here were reported on this cell line, but the second cell line was chosen to be complex enough to provide cell subpopulations with known markers. As we are not aware of a lung cancer cell line with a well-defined intrinsic heterogeneity, we looked for a well-described hematopoietic cell model. Thus, we used U937 where one can obtain a progeny of cells that change their transcriptional program in response to a stimulus (e.g., differentiation) and for which there are established markers (i.e., monocytes and macrophages, see Methods). Please see our response to Reviewer #1, where we explained that U937 were chosen to enable distinguishing cell subpopulations that could be induced from the same/similar progenitor.

This section has been now moved to Methods, following Reviewer # 1 point that this is only technical validation and Reviewer # 2 suggestion "13. Line 170-197, this section has no relationship with the main text, delete or put it into supplementary information." We introduced the U937 cell model in Methods, but just in one sentence, following Reviewer # 2 impression that it may even be deleted: "To test the performance of Drop-seq and computational pipeline Seurat in revealing intrinsic cell subpopulations, the U937 cells were treated with differentiation agent 12-O-tetradecanoylphorbol-13-acetate (TPA) to generate a mixture of monocytes and macrophages".

4. Several cell lines including NSCLC PC9, U937 leukemia cell line and M14 melanoma cell line were used in this study, and these cell lines were investigated in different parts of this paper. Patient survival analysis was only applied in lung cancer.

We have now applied patient survival analysis in melanoma using the markers that we identified in M14 cells treated with vemurafenib. We present analysis of three different melanoma patient data sets in Figure S16C-F. There is no targeted therapy to U937 cells, and this cell line was solely used in the manuscript as a technical control.

Main text: “The markers of DT clusters distinguished patient survival poorly in melanoma, a cancer with much longer survival (Figures S16C-F).”

Figure S16 legend: “(C) Sample level enrichment analysis of DT markers, which were identified in the M14 melanoma cells treated with vemurafenib, was performed in BRAF-mutant TCGA melanomas. Expression level was determined for markers of DT clusters (Clusters 5-8 in Figure 3A and genes in Supplementary Information 7b), for all clusters together (All) and for each individual DT cluster (5,6,7 and 8).

(D) Kaplan-Meier estimates of survival before death/censored in the group of BRAF-mutant melanoma patients (TCGA) with significantly upregulated (Z-score > 1.96, $P < 0.05$) DT markers that were identified in the M14 experiment compared to the group of patients, where DT markers showed decreased expression or no significant change ($P > 0.05$). Survival was calculated based all cluster markers (All) and on the markers of each individual DT cluster (5,6,7 and 8).

(E) and (F) Kaplan-Meier estimates of survival as in (D) performed for GSE65904 and GSE53118 data sets, respectively. Survival was calculated based on the markers of each individual DT cluster (5,6,7 and 8). Patients with upregulated genes (DT markers) survive less in (E), although the P -values of survival difference are not significant.”

Methods: Two melanoma datasets were GSE65904 Illumina Human HT-12V4.0 BeadChip array (<https://www.ncbi.nlm.nih.gov/geo/query/acc.cgi?acc=GSE65904>) and GSE53118 stage III disease using Illumina HumanWG-6 v3.0 BeadChip array (<https://www.ncbi.nlm.nih.gov/geo/query/acc.cgi>), both representing a mix of BRAF- and NRAS-mutant tumors. From the third melanoma dataset, from TCGA SKCM¹⁰³, there was enough BRAF-mutant samples available for analysis, mostly from metastatic disease.

5. Line 18, the key words of ‘single-cell RNA sequencing’ and ‘scRNA-seq’ overlapped.
The keyword “single-cell RNA sequencing” has been removed to keep the abbreviation.

6. Line 100, please clarify how did you test the T790M mutation.
More details are provided in Methods “Mutant allele detection”.

7. Line 112, what does the figure S1C mean?

This was a heatmap showing expression of the top 30 genes in the most significant principal component, PC1, across 100 single cells in our data set. We removed this figure as visualization of expression of the same genes was presented in Figure S1D (now S1E) across all cells.

8. Line 120, the cells might not represent erlotinib-tolerant cells in such a short time of treatment. Test the IC50 of PC9 to erlotinib at different time points.

We tested the IC50 of PC9 that have been treated for 2,4,9, and 11 days and presented the data in comparison to parental PC9 cells in Figure 1C. Even at the earliest time point, two days of preliminary treatment with erlotinib, the cells became much more tolerant and IC50 increased.

9. *Line 129, it is not rational to conclude that the scRNA-seq identified markers of DT cell populations, were associated with epigenetic rather than genetic changes.*

We agree and we have excluded that sentence.

10. *Line 139, there seemed no difference of NEAT1 expression.*

The difference in NEAT1 expression is mostly in the number of small foci, which were visible in the submitted jpg file. The quantitation of the original images in Fig. S2C showed high difference in expression level between treated and untreated cells, where staining was detected in rare cells. We have chosen a different image for erlotinib treatment to show a representative cell that has a larger focus, which hopefully will be visible to the reviewers using the resolution of the merged pdf document.

11. *Line 150, there should be 'five clusters' (4-8) for drug tolerant states of DTPs and DETPs.*

We are grateful to the reviewer for noticing this error. It has been corrected.

12. *Line 152, why do you claim 'it is unlikely that the Cluster 4 cells represent a mixture of sensitive and resistant cells'?*

We have made a smoother transition in the text: "Top Cluster 4 markers were expressed at a lower level in untreated cells, i.e. Clusters, 1, 2, and 3 (Figure 1G); this includes *TACSTD2* that had an increased level in almost every surviving cell at D1 compared to untreated cells (Figure S1E, and Figure S2C and D). These findings make it unlikely that the Cluster 4 represents a mixture of tolerant cells and the cells at the original, sensitive state, and suggest that Cluster 4 cells are rather positioned at a transitional state to DTPs."

13. *Line 170-197, this section has no relationship with the main text, delete or put it into supplementary information.*

Lines 170-197 were transferred to Methods.

14. *Line 284, which figure showed the resistance clusters in PC9 and HCC827 cell lines?*

We have added a UMAP for the PC9 Drop-seq data (Figure 3A). Figure 3B lists drug tolerance (DT) clusters for each cell line.

15. *Line 415, how to define the low/high marker expression?*

Figure 8A shows the range of Z-scores for the dataset. A Z-score between 1.96 and 10 indicates relatively higher (compared to a random set of genes of the same size across the samples) expression level. A Z-score lower than -1.96 indicates relatively lower expression. We have made our description more clear in the text: "We applied sample-level enrichment analysis (SLEA)⁴⁵ to calculate Z-scores for the DT markers. The expression of DT markers was preferentially increased, as the Z-scores were high, in many patients (Figure 8A). Patients with the increased expression of markers of individual DT states or all combined DT markers displayed a decrease in overall survival (Figure 8B and Figure S15A)". Additional description is in the legend to Figure 8A.

16. *Figure 7C, please claim the meanings of 'ASK428, ASK440, ... , EGF^{Rex19}' in the figure legend?*

This has been explained.

17. Line 428, should be 13 different clusters.

“tissues” was changed to “cell types” because we used cell-type specific markers (Table S20) to assign cluster identities.

18. Page 39, in the ‘RT-qPCR’ section, please list all the primers of genes you detected.

This data is now transferred from the Supplementary Notes file to Supplementary Tables file. Reference to Table S25 is given in Methods.

Reviewer #3 (Remarks to the Author):

In this paper, Aissa et al exploit single-cell RNA-sequencing from cell lines as well as from patient-derived samples to explore heterogeneity in lung cancer and show how the results can be used (i) to investigate the mechanisms of drug tolerance and (ii) to design drug combination therapy.

The approach they use is interesting and, in addition to the computational analyses, they validate several of the conclusions that they make.

While there is an impressive amount of datasets being collected, the rationale for including them and the conclusions drawn from the analyses is not always clear. Clarity could be much improved and more insights might even be possible if some of the datasets were better integrated with one another.

We have significantly changed the text in order to integrate data and highlight important findings.

Below my specific comments:

****Major Comments****

1. Line 116-117: "Regressing out cell cycle genes returned similar clusters (data not shown), indicating cell cycle signature did not drive the distribution." This is an interesting piece of analysis that should be reported, as I would imagine that cell cycle might be driving quite a lot of the variability observed at least in the untreated cells. Also in the treated cells, it's not clear why cells from D4, D9 and D11 are sitting between D1 and D2, in the UMAP (Fig 1B); unless this is driven by cell cycle.

We have added data for cell lines treated with erlotinib and crizotinib (Figures S1F and G, and Figure S13B), and for xenografts treated with osimertinib and crizotinib (Figure S14I) where we claim that regressing cell cycle genes does not change attribution of DT states. In fact, in Velocity analysis in Figure 1E, the directionality between cells in Clusters 6,7 and 8 correlates with direction of progression through the cell cycle since cell cycle genes represent the top markers in clusters 7 and 8. There was a single proliferating cluster in the osimertinib-treated xenograft that co-clustered with untreated cells (Cluster 9 in Figure 7C)– after regressing cell cycle genes, this cluster repositioned to the rest of the treated cells (Figure S14I). We have added to the Methods section: "In cell line data, top principal components were dominated by genes involved in DNA synthesis, DNA replication and other processes in the cell cycle. Thus, this

gene set was subtracted before clustering for the testing if cell cycle genes are driving the heterogeneity of different DT states.”

2. *Line 120-121: "From the computational point of view, genes that were not found in the literature appear to be as functionally important as known drug tolerance genes." What does "functionally important" mean here? and how do the authors assess it computationally?*
This sentence was unclear, so it was eliminated.

3. *Line 125-126: isn't the result of the comparison between bulk RNA-seq and Chip-Seq trivial and simply due to the association of H3K4me3 with active genes? Wouldn't DNaseq be necessary to draw the conclusion on Line 128-129 about the epigenetic vs genetic changes (to prove the absence of any DNA mutations)?*
We agree with this opinion. We eliminated that confusing sentence.

4. *Line 155: RNA velocity here could help assessing whether cells in cluster 4 really represent a "transitional state to DTPs".*
We are thankful for this suggestion and now show RNA velocity in Figure 1E. Indeed, it suggest that Cluster 4 transitions to Cluster 5 and 6.

5. *Figure 3A: what's the unit on the y axis? were genes normalized before being compared? if not, this result might be driven by a handful of highly expressed genes*
The data in violin plots are gene expression values. We employed standard pre-processing for all single-cell RNA-seq datasets. Unless otherwise specified, we first performed log-normalization of all datasets, using a size factor of 10,000 molecules for each cell. We next standardized expression values for each gene across all cells (z-score transformation), as it is standard prior to running dimensional reduction tools such as principal component analysis. These steps are implemented in the NormalizeData and ScaleData functions in Seurat.

6. *Figure 3B: a UMAP with cells colored by day of treatment should be added*
The UMAP with color code for day of treatment has been added to that figure (now Figure 4A).

7. *The 10x dataset seems very rich, with multiple cell subpopulations identified, but there's no attempt to interpret those in the paper. Do they represent meaningful cell states or is it just overfitting? How do they compare to the clusters identified in the Dropseq dataset?*
We have reorganized that section of results and discuss different clusters fitting larger subpopulations I and II, as these are relevant to crizotinib tolerance. We are not sure we can extend the paper even further with interpretations of gene signatures in individual clusters.

The authors could try to use one of the many existing algorithms to estimate the optimal number of clusters (eg, gap statistic, silhouette width, etc; this also applies to the other scRNAseq dataset discussed).

We chose to use silhouette width because it is one of the metrics used in Seurat package. We have changed the Methods section: “The Seurat function “FindClusters” was utilized to identify clusters of cells with similar transcriptomes based on their PCs. The dimensions and resolutions were dataset-dependent and are reported in Table S2. The optimal number of clusters were estimated taking in consideration: 1) PCs with p value lower than $1e-10$, calculated using

“JackStraw” function; 2) highest average silhouette width for the combined clusters within a range for resolution from 0.5 to 2; 3) avoiding appearance of different samples in the same cluster; 4) curating clusters for identical top genes (to prevent over-clustering); and 5) the list of enriched gene signatures determined using enrichment analysis, where absent enriched gene signatures would indicate under-clustering, while a heavy overlap in gene signatures between two clusters would indicate over-clustering. The later point is important for revealing non-redundant functional clusters. For the 10x Genomics dataset, in addition to these five criteria, we also used Velocity visualization to point the changes in directionality from one cluster to another and avoid having changes in directionality within one cluster.”

8. How were the two subpopulations of resistant cells (I and II) defined? Was RNA velocity used? It seems particularly difficult to use RNA velocity in this case, as the signal appears fairly weak (Figure 3B, bottom) - or maybe it's a problem of visualization? Projecting the cells on top of eg a PAGA graph (<https://genomebiology.biomedcentral.com/track/pdf/10.1186/s13059-019-1663-x>) would help.

This is right, we used a combination of RNA velocity plotted on UMAP (Figure 4A) and gene signature profile inferred from GiTools enrichment analysis (e.g., Figure 4C). We are thankful to the reviewer for recommending PAGA. We have generated PAGA graph in Figure 4A. Velocity projection showed that the directionality within the tolerant cells goes over cluster 5 and above in direction to clusters 14, 13, 11, 22 forming the subpopulation I; cluster 19 show a directionality going down in direction to clusters 20, 0, 7, 8, 17, and 12 forming the subpopulation II. After projecting the cells on top of a PAGA graph, the subpopulations I and II show high interconnectivity between the clusters within each subpopulation, but less interconnectivity between the two subpopulations.

9. Line 281-288: is this Dropseq data on D3 in different cell lines not shown/discussed? It seems like the only result shown is some enrichment analysis of Figure S8, but if the authors want to use it, they should include further details, as was done for the other scRNAseq datasets (QC, clustering, UMAP/tSNE visualization, etc).

We presented this data in a symmetrical way, for four cell lines, in Figure 3 and Figure S6.

10. Line 290 - 295: which dataset is this referring to? the dropseq dataset mentioned in the paragraph above or some previous dataset?

We transferred this paragraph to relevant data.

11. Line 339-340: are the resistant vs sensitive populations in any way related to the subpopulations I vs II (or any other subpopulation) in the 10x dataset? This is also important for the claim made in lines 366-368.

We changed our text to make it more clear: “the Criz-T markers highly expressed the epithelial G protein-coupled receptor *GPRC5A* and were enriched in gene sets related to epithelium development, drug metabolism, lysosome, and epigenetic signatures, suggesting that Criz-T clusters correspond to the subpopulation II of Erl-treated cells”. The claim in lines 366-368 refers to the lack of new prominent clusters/cell subpopulations after combination treatment compared to erlotinib treatment. We have reached the same conclusion in the new xenograft study.

12. Line 413: "when compared to a random dataset of the same size": what does it mean random? how was it randomized?

We change it to “We applied sample-level enrichment analysis (SLEA)⁴⁵ to calculate Z-scores for the DT markers”, referring to the analysis described in Methods section.

13. Important details are missing/unclear in the Methods section:

a. regarding the QC of the scRNAseq data: which criteria were used to select good/bad quality cells (ie, threshold on number of reads, genes, fraction of reads from mtDNA, etc)?

In Table S2, we present the range of Feature_RNA for inclusion cells in our analysis. For each experiment, we analyzed the number of non-exonic RNA reads, percent.RP, percent.MT, Dispersion to Average Expression, the number of significant PCs, whether there were cells predicted to be doublets, the identity of top cluster markers (such as no irrelevant stress response genes). All these data is available from the corresponding author upon request. For experiments reported in this paper, no criteria were applied to filter cells besides Feature_RNA. We have expanded our explanation on the QC of scRNA-seq in Methods.

b. Line 860-861: which analysis is this referred to? what cell populations were being compared?

We have merged two paragraphs to make sure it is clear that the analysis relates to the information in the previous paragraph.

c. Line 853: "The data were normalized based on total mapped reads in combined analyzed samples". This sentence is unclear. Do the authors mean that the data were normalized by dividing each cell by the total number of reads (ie, RPM normalization)? or else? For RNA-seq analysis, we used the number of mapped reads overlapping a gene and performed differential expression (DE) analysis using R/Bioconductor package DESeq (version 2). DESeq accepts raw counts, not normalized counts. The DESeq is explained in “Comprehensive evaluation of differential gene expression analysis methods for RNA-seq data” (<http://www.genomebiology.com/2013/14/9/R95>): “DESeq computes a scaling factor for a given sample by computing the median of the ratio, for each gene, of its read count over its geometric mean across all samples. It then uses the assumption that most genes are not DE and uses this median of ratios to obtain the scaling factor associated with this sample.”

As described in Methods, we utilized read count reweighting scheme to calculate weighted counts using Genominator (version 1.2.4). Next, the weighted count values were normalized considering two untreated and two treated samples. Using the two biological replicates in each condition, we calculated means of the normalized weighted counts and reported them as baseMean values. All these values are now added to Table S6.

d. Line 875-876: "Drop-seq data was assessed for features of low quality cells of both biological and technical nature". What does this mean exactly? What kind of assessment was performed and what were the results?

The parameters that we used for quality control are now explained in Methods as above in Question 13a. The results for each of the samples that we use in the manuscript are available from the corresponding author upon request.

e. 893: "The resolution was set to highlight the biologically significant differences between cell subpopulations". This is also unclear: which biologically significant differences are the authors referring to, and how were they used to choose the resolution?

We considered avoiding appearance of different samples in the same cluster, and investigated clusters for top genes and gene signatures. Please see the full response to Question 7.

*****Minor Comments*****

** Line 110-111: technically, UMAP is a visualization tool and it shouldn't be used to find cell populations, which are defined by clustering.*

We appreciate this note. It has been corrected.

** Line 167: typo, Figure 2SI -> S2I*

Corrected.

REVIEWERS' COMMENTS

Reviewer #1 (Remarks to the Author):

The authors have mostly addressed my points and the revised manuscript is improved.

Reviewer #2 (Remarks to the Author):

The authors have done a lot of additional work. The additional data from osimertinib will make the study more clinical relevant.

No additional comments.

Thanks for the efforts.

Reviewer #3 (Remarks to the Author):

The authors did a good job addressing all the comments.

The last recommendation I have is to include in the paper all the values of the parameters for the QC, etc (points 13a and 13d in the authors' response). This is key information to allow other people to use the data for reproducing the results or performing additional analyses.

This can easily be done either in the methods or as a supp. table, and there's no reason to make it available "upon request".

Reviewer #1:

The authors have mostly addressed my points and the revised manuscript is improved.

Reviewer #2:

The authors have done a lot of additional work. The additional data from osimertinib will make the study more clinical relevant.

No additional comments.

Thanks for the efforts.

Reviewer #3:

The authors did a good job addressing all the comments.

We thank all reviewers for careful reading of the manuscript and for expressing important points which greatly improved our manuscript.

Reviewer #3:

The last recommendation I have is to include in the paper all the values of the parameters for the QC, etc (points 13a and 13d in the authors' response). This is key information to allow other people to use the data for reproducing the results or performing additional analyses.

This can easily be done either in the methods or as a supp. table, and there's no reason to make it available "upon request".

The distribution of values that were used for the QC are now included within Supplementary Information as Supplementary Fig. 17. The parameters used in each scRNA-seq experiment and manuscript figure are specified in Supplementary Table 2.